# Motor thalamus supports striatum-driven reinforcement

Arnaud L Lalive[1], Anthony D Lien[1], Thomas K Roseberry[1,2†],
Christopher H Donahue[1], Anatol C Kreitzer[1,2,3]*

[1]The Gladstone Institutes, San Francisco, United States; [2]Neuroscience Graduate Program, University of California, San Francisco, United States; [3]Departments of Physiology and Neurology, University of California, San Francisco, United States

**Abstract** Reinforcement has long been thought to require striatal synaptic plasticity. Indeed, direct striatal manipulations such as self-stimulation of direct-pathway projection neurons (dMSNs) are sufficient to induce reinforcement within minutes. However, it's unclear what role, if any, is played by downstream circuitry. Here, we used dMSN self-stimulation in mice as a model for striatum-driven reinforcement and mapped the underlying circuitry across multiple basal ganglia nuclei and output targets. We found that mimicking the effects of dMSN activation on downstream circuitry, through optogenetic suppression of basal ganglia output nucleus substantia nigra reticulata (SNr) or activation of SNr targets in the brainstem or thalamus, was also sufficient to drive rapid reinforcement. Remarkably, silencing motor thalamus—but not other selected targets of SNr—was the only manipulation that reduced dMSN-driven reinforcement. Together, these results point to an unexpected role for basal ganglia output to motor thalamus in striatum-driven reinforcement.

DOI: https://doi.org/10.7554/eLife.34032.001

## Introduction

Reinforcement refers to a process by which the frequency or intensity of a specific behavior increases over time. Standard models of reinforcement invoke dopamine-dependent corticostriatal plasticity as a mechanism underlying this form of associative learning (*Schultz and Dickinson, 2000*; *Reynolds et al., 2001*; *Doya, 2007*; *Maia and Frank, 2011*; *Yagishita et al., 2014*; *Xiong et al., 2015*). In these models, reinforcement occurs through plasticity of excitatory inputs to striatal action-related ensembles, which results in enhanced future recruitment of these motor programs in similar contexts. Yet other studies have found that activating targets of basal ganglia output can also be reinforcing (classic electrical studies reviewed in *Wise, 1996*; for recent optogenetic studies, see below) and reinforcement learning in songbirds and primates is proposed to involve basal ganglia regulation of cortical plasticity (*Turner and Desmurget, 2010*; *Fee and Goldberg, 2011*). Thus, it remains unclear to what extent striatum-driven reinforcement requires engagement of basal ganglia output projections to downstream targets in brainstem or thalamus.

The striatum contains two distinct classes of projection neurons—direct (dMSNs) and indirect pathway medium spiny neurons (iMSNs)—that regulate motivated behavior by increasing or suppressing movement, respectively (*Albin et al., 1989*; *Kravitz et al., 2010*). Numerous lines of evidence support a role for the striatum in reinforcement. Early intracranial self-stimulation experiments showed that striatal stimulation is sufficient to drive operant responding in rodents, in the absence of food reward (*Phillips et al., 1976*; *White and Hiroi, 1998*). In addition, striatal microstimulation biases choice in cue-guided decision tasks in primates (*Nakamura and Hikosaka, 2006*; *Williams and Eskandar, 2006*). Although some of these effects are due to dopamine release from axon stimulation, pathway-specific optogenetic stimulation later revealed that direct pathway

*For correspondence:
akreitzer@gladstone.ucsf.edu

Present address: †Neuralink Corp, San Francisco, United States

Competing interests: The authors declare that no competing interests exist.

stimulation is reinforcing, whereas indirect pathway stimulation drives avoidance (*Hikida et al., 2010*; *Kravitz et al., 2012*). Indeed, mice readily learned an operant task to self-stimulate dMSNs and formed a memory of the stimulation-paired apparatus. This suggests that dMSN stimulation is not just acutely rewarding but can drive long-lasting changes in the brain supporting learning and memory (*Kravitz et al., 2012*). Consistent with this, stimulation of direct and indirect pathways oppositely biases both choice and vigor in water-reinforced tasks (*Tai et al., 2012*; *Yttri and Dudman, 2016*). Similarly, in vivo recordings from primates and rodents show that activity in discrete populations of MSNs encodes reward (*Hikosaka et al., 1989*; *Apicella et al., 1991*; *Oyama et al., 2010*), actions and/or their values (*Lauwereyns et al., 2002*; *Barnes et al., 2005*; *Pasupathy and Miller, 2005*; *Samejima et al., 2005*; *Lau and Glimcher, 2008*; *Kim et al., 2009*; *Kimchi and Laubach, 2009*). Together, these results demonstrate that MSNs both integrate and support motor and reinforcement functions.

The involvement of regions downstream of the basal ganglia in motor control and reinforcement has also been extensively studied. Different brainstem targets of basal ganglia are associated with regulation of distinct aspects motor behavior: the superior colliculus is implicated in orienting (*Hikosaka et al., 2000*), the periaqueductal gray is involved in freezing (*Gross and Canteras, 2012*), and the mesencephalic locomotor region (MLR) is engaged for locomotion (*Esposito and Arber, 2016*), whereas Dorsal Raphe Nucleus (DRN) neurons are involved in mood regulation and state-dependent control of motivation (*Cohen et al., 2015*). Basal-ganglia-recipient motor thalamus, which consists of ventromedial (VM), ventral anterior and lateral (VAL) nuclei, is involved in motor performance and skill learning (*Hikosaka et al., 2000*; *Graybiel, 2005*; *Turner and Desmurget, 2010*; *Bosch-Bouju et al., 2013*; *Goldberg et al., 2013*; *Kawai et al., 2015*). Extensive intracranial self-stimulation studies have revealed that a multitude of sites across the brain, including different brainstem targets of basal ganglia, are sufficient to drive reinforcement (*Wise, 1996*). Recent optogenetic studies also showed that activation of specific cell populations in the MLR (*Dautan et al., 2016*; *Xiao et al., 2016*; *Yoo et al., 2017*) and DRN (*Liu et al., 2014*; *McDevitt et al., 2014*) is reinforcing. Additionally, electrical stimulation of motor thalamus (*Clavier and Gerfen, 1982*) supports operant behavior, and lesions impair striatum-dependent learning (*Zis et al., 1984*). However, it is poorly understood whether these basal ganglia output targets are necessary for striatum-driven reinforcement. Indeed, after almost 70 years of research on the anatomical substrates of reinforcement, it is still debated whether nuclei supporting self-stimulation operate in series within a single circuit, or belong to multiple, parallel systems (*Routtenberg, 1971*; *Phillips, 1984*; *Wise and Bozarth, 1984*).

Here, we took advantage of optogenetic dMSN self-stimulation in mice to probe both striatal plasticity and basal ganglia circuit mechanisms in striatum-driven reinforcement. In theory, repeated pairing of operant behavior with dMSN depolarization could locally induce Hebbian synaptic plasticity onto stimulated dMSNs and/or engage downstream circuits to drive reinforcement. We found that dMSN self-stimulation was not accompanied by measurable changes in synaptic strength of excitatory afferents, nor was it impaired by disrupting NMDA receptors in stimulated neurons. Furthermore, reinforcement could be elicited by direct suppression of SNr. In order to identify the contribution of basal ganglia target regions to striatal reinforcement, we combined dMSN self-stimulation with silencing of selected basal ganglia output targets. Although brainstem and thalamic targets were sufficient to drive reinforcement, only motor thalamus was necessary for dMSN self-stimulation. Thus, our results highlight an underappreciated circuit mechanism for striatum-driven reinforcement through recruitment of motor thalamus.

## Results

### dMSN self-stimulation supports reinforcement independently of NMDA receptors

We bilaterally injected Cre-dependent ChR2-eYFP virus and implanted optical fibers in the dorsomedial striatum (DMS) of D1-Cre mice (D1-ChR2) for selective self-stimulation of dMSNs, which represented the primary reinforcer motivating task performance (*Figure 1a*, *Figure 1—figure supplement 1a*). Mice learned to self-stimulate dMSNs by poking in an active (laser-paired) nosepoke (*Figure 1b*), whereas a second, inactive nosepoke had no effect. We characterized the self-

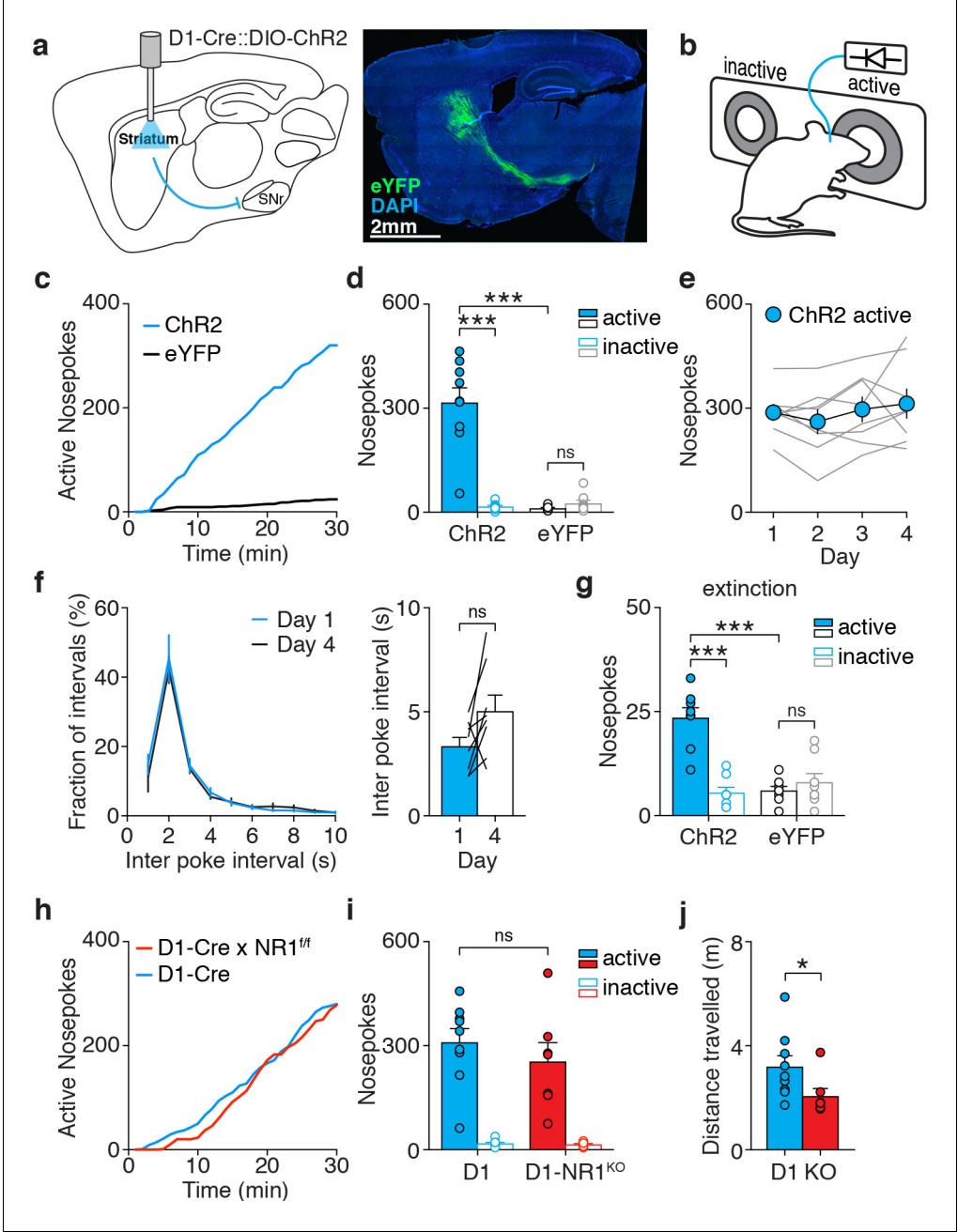

**Figure 1.** dMSN self-stimulation elicits rapid and persistent NMDAR-independent reinforcement. (**a**) Schematic of a sagittal brain section showing optogenetic stimulation of striatal dMSNs (left), and slice showing fluorescent dMSNs after striatal infusion of a Cre-dependent virus encoding eYFP in a D1-Cre mouse (right). (**b**) Schematic of the behavioral apparatus showing active (laser-paired) and inactive nosepokes. (**c**) Example cumulative plot of active nosepokes from a D1-ChR2 (blue) and D1-eYFP (black) mouse. (**d**) Average nosepokes during a single self-stimulation session for D1-ChR2 and D1-eYFP mice (n = 11 and 8, two-way ANOVA, interaction poke x group, $F_{(1,17)}$=31.03, p<0.001, posthoc Sidak's multiple comparisons test, ChR2 active vs eYFP active p<0.001, ChR2 active vs inactive p<0.001, eYFP active vs inactive p=0.921). (**e**) Average nosepokes during four consecutive daily sessions (n = 8, one-way ANOVA, $F_{(1.908,13.36)}$ = 1.237, p=0.319). (**f**) Distribution of (left) and average (right) inter-poke intervals for days 1 and 4 of dMSN self-stimulation (n = 8, paired t-test, p=0.0575). (**g**) Average nosepokes during an extinction session for D1-ChR2 and D1-eYFP mice (n = 8 and 8, two-way ANOVA, interaction poke x group, $F_{(1,14)}$=47.86, p<0.0001, posthoc Sidak's multiple comparisons test, ChR2 active vs eYFP active p<0.0001, ChR2 active vs inactive p<0.0001, eYFP active vs inactive p=0.5703). (**h**) Example cumulative plot of active nosepokes from D1-ChR2 mice expressing (D1-Cre, blue) or lacking (D1-Cre x NR1f/f, red) NMDA receptors in dMSNs. (**i**)

*Figure 1 continued on next page*

*Figure 1 continued*

Average nosepokes during a single self-stimulation session for D1-ChR2 mice with ('D1') or without ('D1-NR1$^{KO}$') NMDA receptors in dMSNs (n = 9 and 7, two-way ANOVA, interaction poke x group, F(1,14)=0.638, p=0.438, posthoc Sidak's multiple comparisons test, D1 active vs D1-NR1 active p=0.418). (j) Average distance travelled by D1-ChR2 mice with ('D1') or without ('KO') NMDA receptors in dMSNs (Mann Whitney U test, p=0.023).
DOI: https://doi.org/10.7554/eLife.34032.002

The following source data and figure supplements are available for figure 1:

**Source data 1.** Source data for *Figure 1*.
DOI: https://doi.org/10.7554/eLife.34032.008
**Figure supplement 1.** Fiber placement and infusion sites.
DOI: https://doi.org/10.7554/eLife.34032.003
**Figure supplement 2.** Characterization of dMSN self-stimulation.
DOI: https://doi.org/10.7554/eLife.34032.004
**Figure supplement 2—source data 1.** Source data for *Figure 1—figure supplement 2*.
DOI: https://doi.org/10.7554/eLife.34032.005
**Figure supplement 3.** dMSN self-stimulation pattern is stable over days.
DOI: https://doi.org/10.7554/eLife.34032.006
**Figure supplement 3—source data 1.** Source data for *Figure 1—figure supplement 3*.
DOI: https://doi.org/10.7554/eLife.34032.007

stimulation behavior by varying different task parameters. Poke rate was sensitive to laser duration, stimulation pattern, effort, and contingency degradation, all of which are hallmarks of goal-directed behavior (*Figure 1—figure supplement 2*). For subsequent experiments, we chose one second of constant laser stimulation per nosepoke to optimally balance poke rate and total laser stimulation. Within a single session, D1-ChR2 mice began to make more nosepokes on the active side and rapidly neglected the inactive nosepoke (*Figure 1c,d*). In contrast, control mice (expressing only eYFP) produced fewer and equal numbers of nosepokes on both sides. Overall, no difference in total locomotion was observed during the session (*Figure 1—figure supplement 2f*). The total number of active nosepokes per session was stable over days (*Figure 1e*). Similarly, poking patterns did not change over time, showing stable inter-poke interval distributions and averages between day 1 and 4. Poking behavior could be divided in epochs of poking bursts, defined here by at least two nosepokes performed in <2 s, or an alternative criterion of <6 s. In either case, the number of bursts and pokes/bursts did not change over days (*Figure 1—figure supplement 3*). Additionally, when mice were subjected to an extinction session 24 hr after the last training session, D1-ChR2 mice made significantly more nosepokes on the previously-active side, despite the absence of dMSN stimulation (*Figure 1g*). In contrast, D1-eYFP mice showed no preference for either side. Together these results demonstrate that behavior was learned and stabilized within a single session, and that mice form a stable association (lasting at least 24 hr) between the nosepoke and the reinforcer (dMSN stimulation). Therefore, we focused our analysis on the first behavior session (see materials and methods).

Current models of basal ganglia reinforcement posit that synaptic plasticity of excitatory cortical inputs onto MSNs supports learning (*Schultz and Dickinson, 2000*; *Doya, 2007*; *Maia and Frank, 2011*). NMDA receptors have been implicated in the induction of various forms of synaptic plasticity in the striatum (*Calabresi et al., 1992*; *Thomas et al., 2000*; *Shen et al., 2008*), and interfering with striatal NMDA receptors disrupts action-outcome associations (*Yin et al., 2005a*). To assess a role for synaptic plasticity in dMSN reinforcement, we generated mice lacking NMDA receptors in dMSNs (D1-Cre x NR1$^{f/f}$), in which various forms of plasticity are altered (*Dang et al., 2006*). However, these mice showed normal dMSN self-stimulation, despite a mild decrease in locomotion (*Figure 1h–j*, *Figure 1—figure supplement 1a*).

We next tested whether any trace of synaptic plasticity could be observed in stimulated (ChR2+) dMSNs after training. We trained D1-Cre x D1-tmt mice injected with Cre-dependent ChR2 or eYFP and then prepared acute slices for whole-cell recordings. We targeted neurons directly underneath the optic fiber tip (*Figure 2a*). We recorded: ChR2 +and ChR2- dMSNs (identified by tmt expression and presence or absence of ChR2-dependent photocurrent, respectively) in D1-ChR2 mice; eYFP +neurons in D1-eYFP mice, and tmt +dMSNs in naive D1-tmt mice. We examined mEPSCs, AMPA/NMDA ratios, paired-pulse ratio, and intrinsic excitability of dMSNs, but found no difference

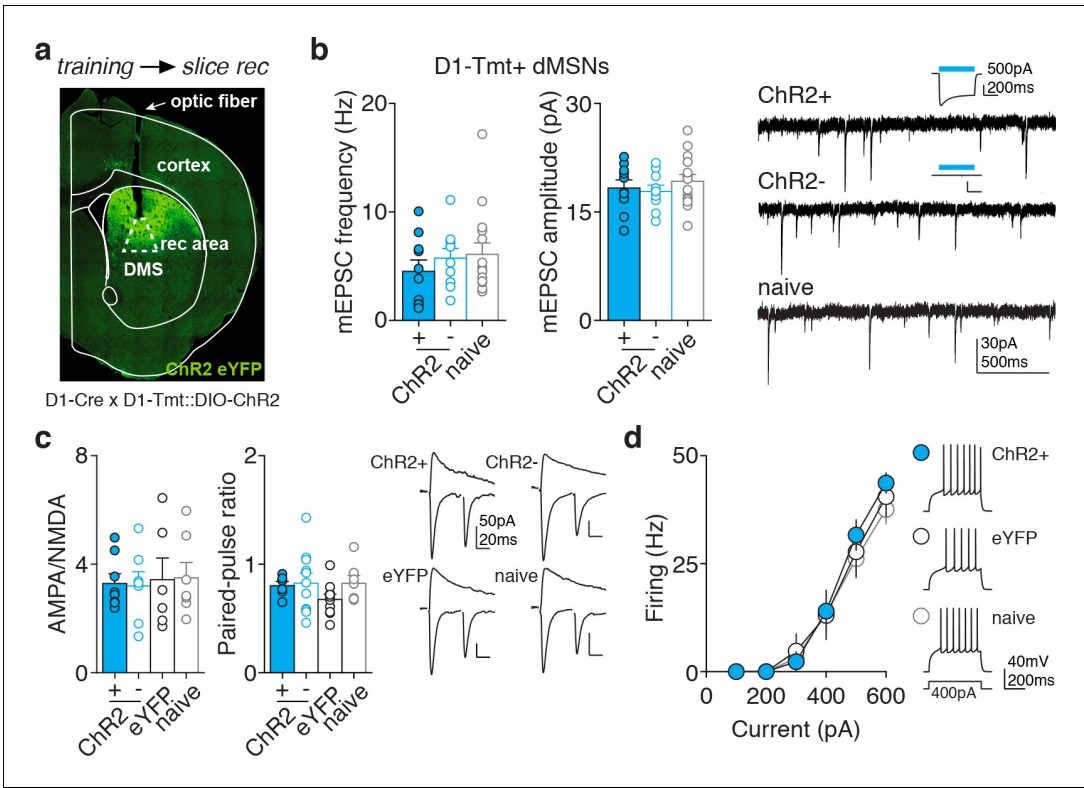

**Figure 2.** No measurable changes in excitatory synaptic transmission or excitability in dMSNs after self-stimulation. (a) Coronal section of the striatum showing ChR2-eYFP expression in DMS, optic fiber track and recording area (dotted box), corresponding approximately to the portion of striatum illuminated during behavior. (b) Whole-cell recordings of dMSNs (tmt+) from trained D1-ChR2 mice (including ChR2 +and ChR2- neurons, identified by the presence or absence of a blue light-evoked photocurrent, inset) and naive mice (tmt+), showing average (left) and example traces (right) for mEPSC frequency and amplitude (frequency: Kruskal-Wallis test, p=0.505; amplitude, one-way ANOVA, $F_{(2,32)}=0.6141$, p=0.547). (c,d) Whole-cell recordings of dMSNs (tmt+) from trained D1-ChR2 mice (including ChR2 +and ChR2- neurons), trained D1-eYFP mice (eYFP +neurons), or naive mice (tmt+), showing averages (left) and example traces (right) for AMPA/NMDA ratio and paired-pulse ratio (c) and excitability (d) [AMPA/NMDA ratio: n = 8,7,6,7; one-way ANOVA, $F_{(3,24)}=0.062$, p=0.9793; paired-pulse ratio: n = 8,11,11,7; one-way ANOVA, $F_{(3,32)}=1.358$, p=0.273; excitability: n = 6,8,14; two-way ANOVA, interaction current x group $F_{(10,125)}=0.3801$, p=0.9533].

DOI: https://doi.org/10.7554/eLife.34032.009

The following source data and figure supplements are available for figure 2:

**Source data 1.** Source data for *Figure 2*.
DOI: https://doi.org/10.7554/eLife.34032.012

**Figure supplement 1.** No measurable changes in excitatory synaptic transmission or excitability in iMSNs after dMSN self-stimulation.
DOI: https://doi.org/10.7554/eLife.34032.010

**Figure supplement 1—source data 1.** Source data for *Figure 2—figure supplement 1*.
DOI: https://doi.org/10.7554/eLife.34032.011

in any of these parameters after training (*Figure 2*). Similarly, we probed synaptic transmission and excitability in iMSNs (D1-tmt negative) after dMSN self-stimulation and did not observe any plastic changes (*Figure 2—figure supplement 1*). Taken together, these results prompted us to explore the involvement of circuitry downstream of striatum in reinforcement.

## Reinforcement through modulation of circuitry downstream of striatum

Given the widespread influence of the basal ganglia over multiple brain structures (*Freeze et al., 2013*; *Oldenburg and Sabatini, 2015*; *Lee et al., 2016*), we asked which specific downstream

circuits could be involved in dMSN-driven reinforcement. Activation of inhibitory dMSNs leads to a net suppression of SNr neurons and disinhibition of downstream targets. First, we tested whether bypassing the striatum and directly inhibiting the SNr would be sufficient to recapitulate direct pathway-driven reinforcement. To do so, we bilaterally expressed Cre-dependent Arch3.0 in the SNr of vGAT-Cre mice. Surprisingly, these animals also learned to self-inhibit the SNr (*Figure 3a–c*, *Figure 1—figure supplement 1b*).

The SNr distributes basal ganglia output across several brain regions, including a canonical projection to ventromedial thalamus, traditionally recognized as part of the motor thalamus (*Starr and Summerhayes, 1983*; *Klockgether et al., 1985*; *Deniau and Chevalier, 1992*). It was recently shown that VM neurons send projections to and interact with prefrontal cortices, which are involved in decision-making and learning (*Kuramoto et al., 2015*; *Guo et al., 2017*). Therefore, we asked whether activation of VM would be sufficient to support operant behavior. We targeted VM neurons with CaMKIIα-ChR2-containing virus in WT mice. These mice showed robust self-stimulation of VM (*Figure 3d–f*, *Figure 1—figure supplement 1c*).

Brainstem targets of SNr, including the DRN (*Liu et al., 2014*; *McDevitt et al., 2014*; *Cohen et al., 2015*) and the pedunculopontine tegmentum, a part of the MLR (*Xiao et al., 2016*; *Yoo et al., 2017*), have also been associated with reinforcement. However, there is some uncertainty about the neuronal identity of cell populations involved. When targeting serotonergic neurons of the DRN, we observed self-stimulation in SERT-Cre mice injected with Cre-dependent ChR2 (*Figure 3g–i*, *Figure 1—figure supplement 1d*), but not in ePET-Cre mice (data not shown). In the MLR, glutamatergic neurons have been shown to receive innervation from SNr and drive both locomotion (*Roseberry et al., 2016*) and reinforcement (*Yoo et al., 2017*). We targeted these neurons with Cre-dependent ChR2 infusion into the MLR of vGLUT2-Cre mice and functionally validated expression by confirming increased locomotion upon stimulation (data not shown). When placed into the operant chamber, 2 of 5 mice showed clear self-stimulation behavior, whereas the other three showed no preference (*Figure 3j–l*, see materials and methods for responder criterion). Taken together, these results show that self-stimulation behavior can be elicited in both the DRN and MLR, raising the possibility that they are also engaged during dMSN self-stimulation to mediate reinforcement.

## DRN serotonergic and MLR glutamatergic neurons are not required for dMSN-driven reinforcement

We next asked whether activation of DRN serotonergic neurons or MLR glutamatergic neurons is necessary for dMSN-driven reinforcement. To do so we combined dMSN self-stimulation with various approaches to silence these downstream nuclei. To test the role of DRN serotonin neurons, we virally expressed Cre-dependent Caspase three in SERT-Cre mice, leading to loss of SERT + cells (*Figure 1—figure supplement 1e*). Selective dMSN excitation was achieved by injecting hSyn-ChR2-eYFP virus into the striatum and placing optic fibers above the anterior tip of SNr, where only fibers from dMSNs are present (*Figure 4a*). This approach yielded reliable dMSN self-stimulation, similar to striatal cell body self-stimulation (*Figure 1—figure supplement 2e*). However, we observed no difference in the number of nosepokes performed by mice with lesioned or intact DRN (*Figure 4b, c*).

We then asked whether glutamatergic neurons in the MLR could be involved in dMSN-driven reinforcement. We expressed Cre-dependent ChR2 in the striatum of D1-Cre mice and eNpHR3.0 in the MLR under the CaMKIIα promoter to target glutamatergic neurons (*Figure 4d*). We confirmed that silencing MLR glutamatergic neurons reduced dMSN-evoked locomotion (*Figure 4—figure supplement 1*), as previously reported (*Roseberry et al., 2016*). However, we observed no difference in the total number of nosepokes in mice for dMSN stimulation alone or paired with inhibition of MLR glutamatergic neurons (*Figure 4e,f*).

## VM supports dMSN-driven reinforcement

Given that the brainstem targets of basal ganglia output that we tested do not appear to be required for dMSN-mediated reinforcement, we next examined the role of SNr-recipient VM thalamus. In order to test whether VM activity contributes to dMSN-mediated reinforcement, we inhibited VM with muscimol, a GABA$_A$ receptor agonist, in D1-ChR2 mice prior to optogenetic self-stimulation (*Figure 5a*, *Figure 1—figure supplement 1f*). Overall, VM-muscimol mice self-stimulated less than

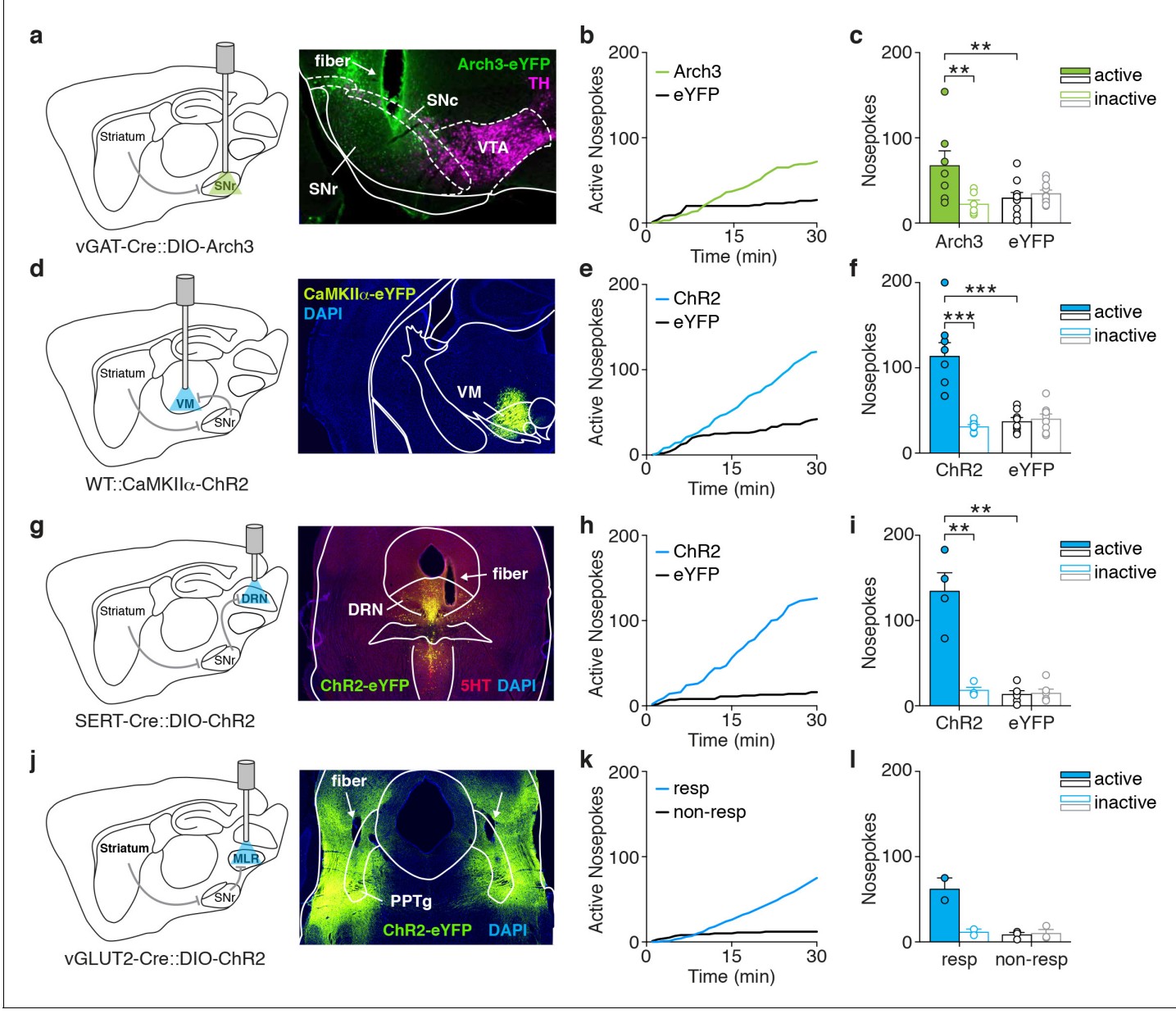

**Figure 3.** Optogenetic control of basal ganglia output neurons or their projection targets is reinforcing. (a) Schematic of a sagittal brain section showing optogenetic inhibition of SNr (left), and coronal slice showing Arch3-eYFP and TH expression after SNr infusion of a Cre-dependent construct in a vGAT-Cre mouse and fiber track (right). (b) Example cumulative plot of active nosepokes from a SNr-Arch3 (green) and SNr-eYFP (black) mouse. (c) Average nosepokes during a single self-inhibition session for SNr-Arch3 and SNr-eYFP mice (n = 7 and 10, two-way ANOVA, interaction poke x group, F(1,15)=13.01, p=0.003, posthoc Sidak's multiple comparisons test, Arch3 active vs eYFP active p=0.007). (d) Schematic of a sagittal brain section showing optogenetic excitation of VM (left), and coronal slice showing ChR2-eYFP expression after VM infusion of CaMKIIα-ChR2 in a WT mouse (right). (e) Example cumulative plot of active nosepokes from a VM-ChR2 (blue) and VM-eYFP (black) mouse. (f) Average nosepokes during a single self-stimulation session for VM-ChR2 and VM-eYFP mice (n = 8 and 8, two-way ANOVA, interaction poke x group, F(1,14)=22.17, p=0.003, posthoc Sidak's multiple comparisons test, ChR2 active vs eYFP active p<0.001). (g) Schematic of a sagittal brain section showing optogenetic excitation of DRN (left), and coronal slice showing ChR2-eYFP overlapping with 5HT expression after infusion of DIO-ChR2 in a SERT-Cre mouse and fiber track (right). (h) Example cumulative plot of active nosepokes from a DRN-ChR2 (blue) and DRN-eYFP (black) mouse. (i) Average nosepokes during a single self-stimulation session for DRN-ChR2 and DRN-eYFP mice (n = 4 and 6, two-way ANOVA, interaction poke x group, F(1,8)=38.09, p<0.001, posthoc Sidak's multiple comparisons test, ChR2 active vs eYFP active p<0.001). (j) Schematic of a sagittal brain section showing optogenetic excitation of the MLR (left), and coronal slice showing ChR2-eYFP expression after MLR infusion (centered around the PPTg) of DIO-ChR2 in a vGLUT2-Cre mouse and fiber tracks (right). (k) Example cumulative plot of active nosepokes from an MLR-ChR2 responsive (resp, blue) and MLR-ChR2 non-responsive (non-resp, black) mouse. (l) Average nosepokes during a single self-stimulation session for MLR-ChR2 responsive and non-responsive mice (n = 2 and 3).

*Figure 3 continued on next page*

*Figure 3 continued*

Abbreviations: 5HT, 5-hydroxytryptamine; DRN, dorsal raphe nucleus; MLR, mesencephalic locomotor region, PPTg, pedunculopontine tegmentum; SNc, substantia nigra compacta; SNr, substantia nigra reticulata; TH, tyrosine hydroxylase; VM, ventromedial thalamus; VTA, ventral tegmental area.
DOI: https://doi.org/10.7554/eLife.34032.013

The following source data is available for figure 3:

**Source data 1.** Source data for *Figure 3*.
DOI: https://doi.org/10.7554/eLife.34032.014

saline-infused control mice (*Figure 5b,c*). In contrast, VM silencing had no effect on either spontaneous or dMSN-evoked locomotion (*Figure 5d,e*). To examine the requirement for VM with higher temporal resolution, we injected Cre-dependent ChR2-eYFP in the striatum and DIO-ChR2-eYFP or DIO-eYFP control virus in the SNr of vGAT-Cre mice (*Figure 5f*, *Figure 1—figure supplement 1g*). For dMSN-selective stimulation, we bilaterally implanted optic fibers above the cerebral peduncle, which contains axons of dMSNs on their way to the SNr. We also implanted a second set of fibers

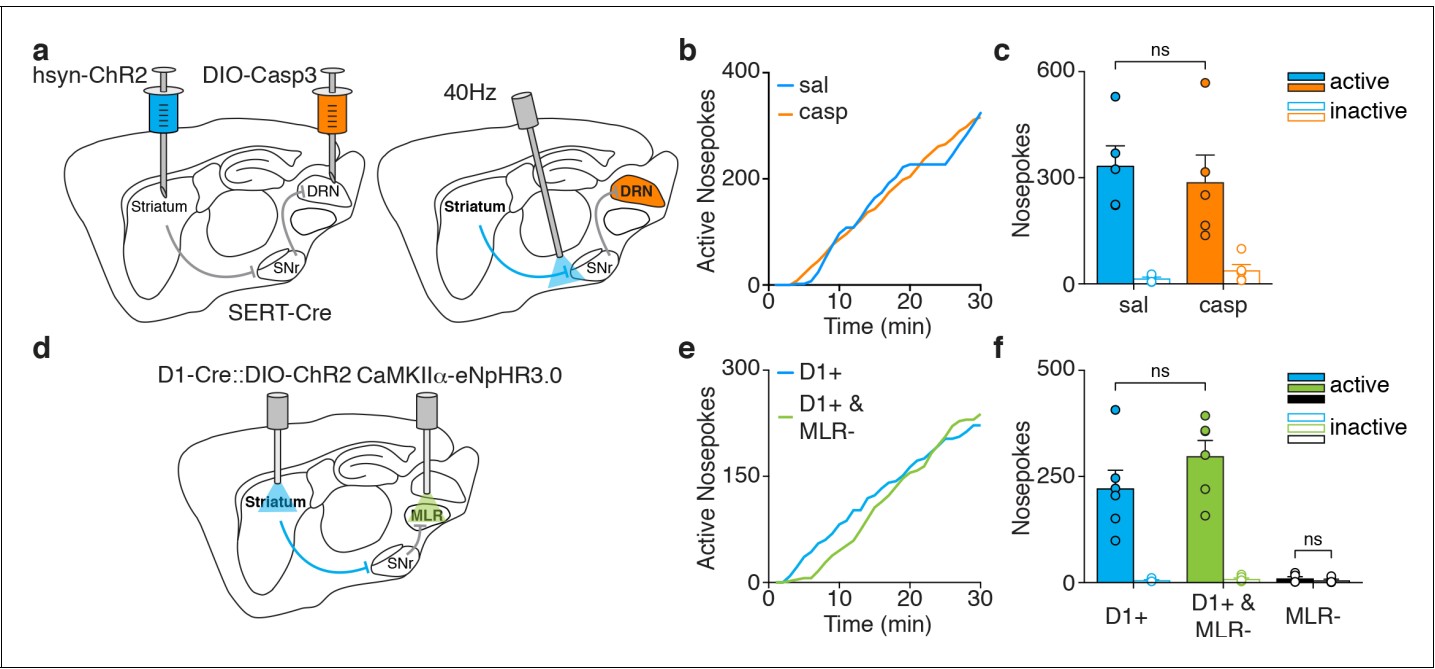

**Figure 4.** Silencing DRN serotonergic neurons or MLR glutamatergic neurons does not disrupt dMSN self-stimulation. (**a**) Sagittal schematic showing injection of a hSyn-ChR2 construct in the striatum and DIO-Casp3 in the DRN of a SERT-Cre mouse, and optic fiber placement above dMSN axons. (**b**) Example cumulative plot of active nosepokes for dMSN axon stimulation in mice infused with saline (sal, blue) or DIO-Casp3 (casp, orange). (**c**) Average nosepokes during a single self-stimulation session for dMSN axon self-stimulation in mice with intact (sal) or lesioned (casp) serotonergic neurons in DRN (sal versus casp, two-way ANOVA, interaction pokes x stim $F_{(1,8)}$=0.45, p=0.521, posthoc Sidak's multiple comparisons test, sal active vs casp active p=0.76). (**d**) Sagittal schematic showing optogenetic stimulation of dMSNs and inhibition of MLR glutamatergic neurons (CaMKIIα-eNpHR3.0). (**e**) Example cumulative plot of active nosepokes from D1-ChR2 mice for dMSN stimulation alone (blue) or paired with MLR inhibition (green). (**f**) Average nosepokes during a single self-stimulation session for dMSN stimulation alone or paired with MLR inhibition, or MLR inhibition alone (D1 +versus D1+ and MLR-, two-way ANOVA, interaction pokes x stim $F_{(1,10)}$=1.618, p=0.232, posthoc Sidak's multiple comparisons test, D1 +active vs D1+ and MLR-active p=0.142; MLR-, Wilcoxon signed rank test, active vs inactive p=0.625).
DOI: https://doi.org/10.7554/eLife.34032.015

The following source data and figure supplements are available for figure 4:

**Source data 1.** Source data for *Figure 4*.
DOI: https://doi.org/10.7554/eLife.34032.018

**Figure supplement 1.** MLR glutamatergic neurons are necessary for dMSN-driven locomotion.
DOI: https://doi.org/10.7554/eLife.34032.016

**Figure supplement 1—source data 1.** Source data for *Figure 4—figure supplement 1*.
DOI: https://doi.org/10.7554/eLife.34032.017

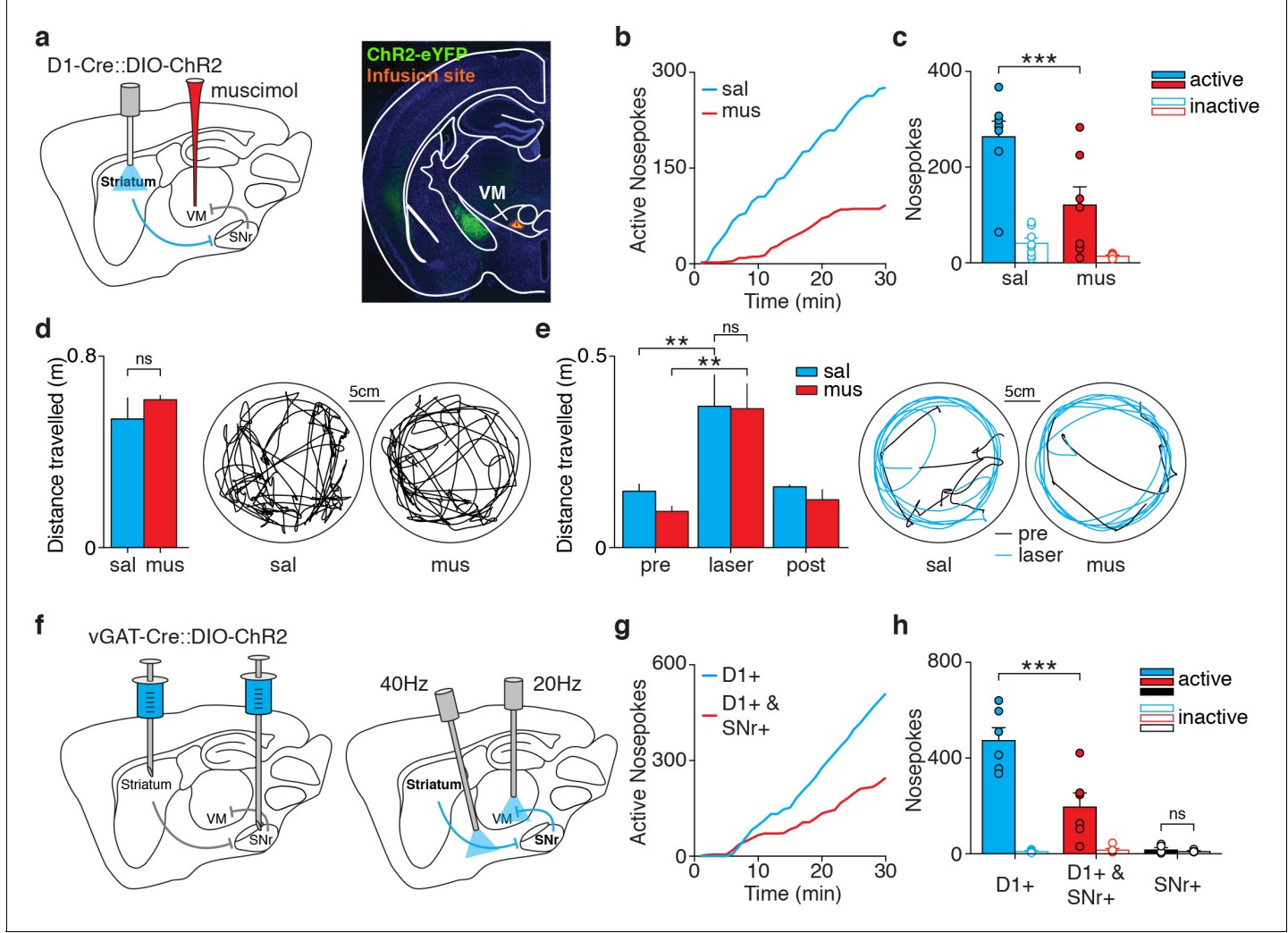

**Figure 5.** VM thalamus inhibition disrupts dMSN self-stimulation. (**a**) Schematic of a sagittal brain section showing optogenetic excitation of striatal dMSNs combined with muscimol infusion in VM, and coronal slice from a D1-Cre mouse showing ChR2-eYFP-expressing fibers from dMSNs en route to SNr (green) and infusion site in VM (DiI, orange). (**b**) Example cumulative plot of active nosepokes from D1-ChR2 mice infused with saline (sal, blue) or muscimol (mus, red). (**c**) Average nosepokes during a single self-stimulation session for D1-ChR2 mice infused with saline (sal) or muscimol (mus, n = 8 and 7, two-way ANOVA, interaction poke x group, $F_{(1,13)}=6.33$, p=0.026, posthoc Sidak's multiple comparisons test, sal active vs mus active p=0.0009, sal active vs inactive p<0.001, musactive vs inactive p=0.018). (**d**) Average spontaneous locomotion in D1-ChR2 mice infused in VM with muscimol or saline (left), and example tracks (right) (n = 3, Wilcoxon matched-pairs signed rank test, p=0.5). (**e**) Average distance travelled before (pre), during (laser) and after (post) dMSN stimulation in the same mice as in g (left), and example tracks (right) (two-way ANOVA, interaction laser x drug $F_{(2,4)}=0.873$, p=0.485, main effect of laser $F_{(2,4)}=14.16$, p=0.015, posthoc Tukey's multiple comparisons test, sal pre vs laser p=0.002, mus pre vs laser p=0.001, sal laser vs mus laser p=0.994). (**f**) Schematic of a sagittal brain section showing injection of Cre-dependent ChR2 in the striatum and SNr of a vGAT-Cre mouse (left) and fiber placement above dMSN axons and in VM for optogenetic excitation of dMSN combined with excitation of SNr terminals in VM, respectively (right). (**g**) Example cumulative plot of active nosepokes for dMSN axon stimulation paired with SNr-eYFP (D1+, blue) or SNr-ChR2 terminal stimulation in VM (D1+ and SNr+, red). (**h**) Average nosepokes during a single self-stimulation session from the same mice as in e (n = 6 and 6, two-way ANOVA, interaction poke x group, $F_{(1,10)}=12.95$, p=0.0049, posthoc Sidak's multiple comparisons test, D1 +active vs D1+ and SNr +active p<0.001, D1 +active vs inactive p<0.001, D1+ and SNr +active vs inactive p=0.018; SNr+, Wilcoxon signed rank test, active vs inactive p=0.625).

DOI: https://doi.org/10.7554/eLife.34032.019

The following source data is available for figure 5:

**Source data 1.** Source data for *Figure 5*.
DOI: https://doi.org/10.7554/eLife.34032.020

above VM, to specifically manipulate axon terminals of VM-projecting SNr neurons. This design enabled us to prevent VM disinhibition only during dMSN stimulation. Similar to what we observed with muscimol infusions, mice made fewer nosepokes when dMSN stimulation was paired with SNr terminal stimulation in VM, compared to control mice expressing only eYFP in the SNr (*Figure 5g,h*). SNr terminals were stimulated at 20 Hz, a frequency similar to basal SNr firing rates in mice (*Freeze et al., 2013*), and which alone was not reinforcing. Together, these results indicate that VM disinhibition is a critical component of the reinforcing circuit initiated by dMSN self-stimulation.

Finally, to better understand how basal ganglia output modulates VM, we performed ex vivo and in vivo recordings in VM during stimulation of basal ganglia circuitry. We first expressed ChR2 in the SNr of vGAT-Cre mice. We prepared acute slices of VM from the same mice and performed whole-cell recordings in VM that confirmed the inhibitory nature of this projection (*Figure 6—figure supplement 1a*). Next, we performed extracellular recordings from 163 VM neurons in awake, head-fixed mice while stimulating dMSNs for 1 s, mimicking task conditions (*Figure 6a*, *Figure 6—figure supplement 1b*). VM cells exhibited a variety of responses, from excitation to inhibition (*Figure 6b*), with the average Z-score of all units showing a net increase in firing during dMSN stimulation (*Figure 6c*). Within 20 ms of stimulation, most responsive VM neurons were excited (*Figure 6d*); these excited neurons displayed a shorter latency to change firing rate than inhibited cells. Within a 100 ms of stimulation, more cells were excited, and a significant fraction of neurons with decreased activity appeared (*Figure 6e*). Over 1 s of stimulation, a majority of neurons were excited, while a small fraction of cells did not show any significant change in firing (*Figure 6f*). There was no obvious spatial segregation of response type within VM (*Figure 6—figure supplement 1c*). These results demonstrate that dMSN activation increases activity in a majority of VM thalamus neurons and, taken together with the VM self-stimulation data (*Figure 3*) and inactivation data (*Figure 5*), further support a role for VM disinhibition in dMSN-driven reinforcement.

## Discussion

We used striatal dMSN self-stimulation as a model to dissect the circuit mechanisms underlying a simple form of striatum-driven reinforcement. Our results highlight several important aspects of basal ganglia function. First, we demonstrate that activation of striatal direct pathway projection neurons can engage a form of reinforcement that is independent of NMDAR-dependent plasticity in dMSNs. Second, we find that multiple downstream targets of basal ganglia output can mediate reinforcement, not all of which are required for dMSN self-stimulation. Finally, we identify an unexpected role for motor thalamus in supporting dMSN-driven reinforcement.

### An alternative to classic mechanisms of reinforcement

Models of the basal ganglia typically propose that corticostriatal synaptic plasticity underlies reinforcement (*Schultz and Dickinson, 2000*; *Doya, 2007*; *Maia and Frank, 2011*). We observed here that dMSN self-stimulation plateaued during the first day of training and that mice formed a memory of the dMSN stimulation-paired nosepoke, consistent with our previous report (*Kravitz et al., 2012*). These results suggest that behavior was learned and acquired within a single session, presumably by engaging long-term plasticity. We initially hypothesized that the repeated pairing of a specific action (entry into the active nosepoke) with depolarization of ChR2-expressing dMSNs would drive NMDAR-dependent Hebbian plasticity at active (task-related) synapses onto stimulated dMSNs. This mechanism would thereby provide a synaptic substrate for reinforcement learning, through the task-dependent recruitment of action-promoting dMSNs. However, we observed that dMSN self-stimulation was unaltered by NMDAR knockout in dMSNs (*Figure 1*). Additionally, despite our ability to identify and record from task-relevant ChR2-expressing dMSNs, we did not observe changes in pre or postsynaptic excitatory transmission, or in neuronal excitability, across both dMSNs and iMSNs (*Figure 2*). In fact, previous reports suggest that simple types of reinforcement can occur without NMDAR-dependent striatal plasticity. For example, mice with dorsal striatum lesions can still learn and perform an action to obtain food rewards (*Yin et al., 2005b*). Similarly, altering MSN NMDAR composition by knocking out GluN2B, an important subunit for plasticity induction (*Paoletti et al., 2013*), does not prevent mice from learning a basic operant task (*Brigman et al., 2013*). Likewise, genetic deletion of NMDA receptors in MSNs does not prevent mice from learning simple action-outcome associations (*Jin and Costa, 2010*); although see

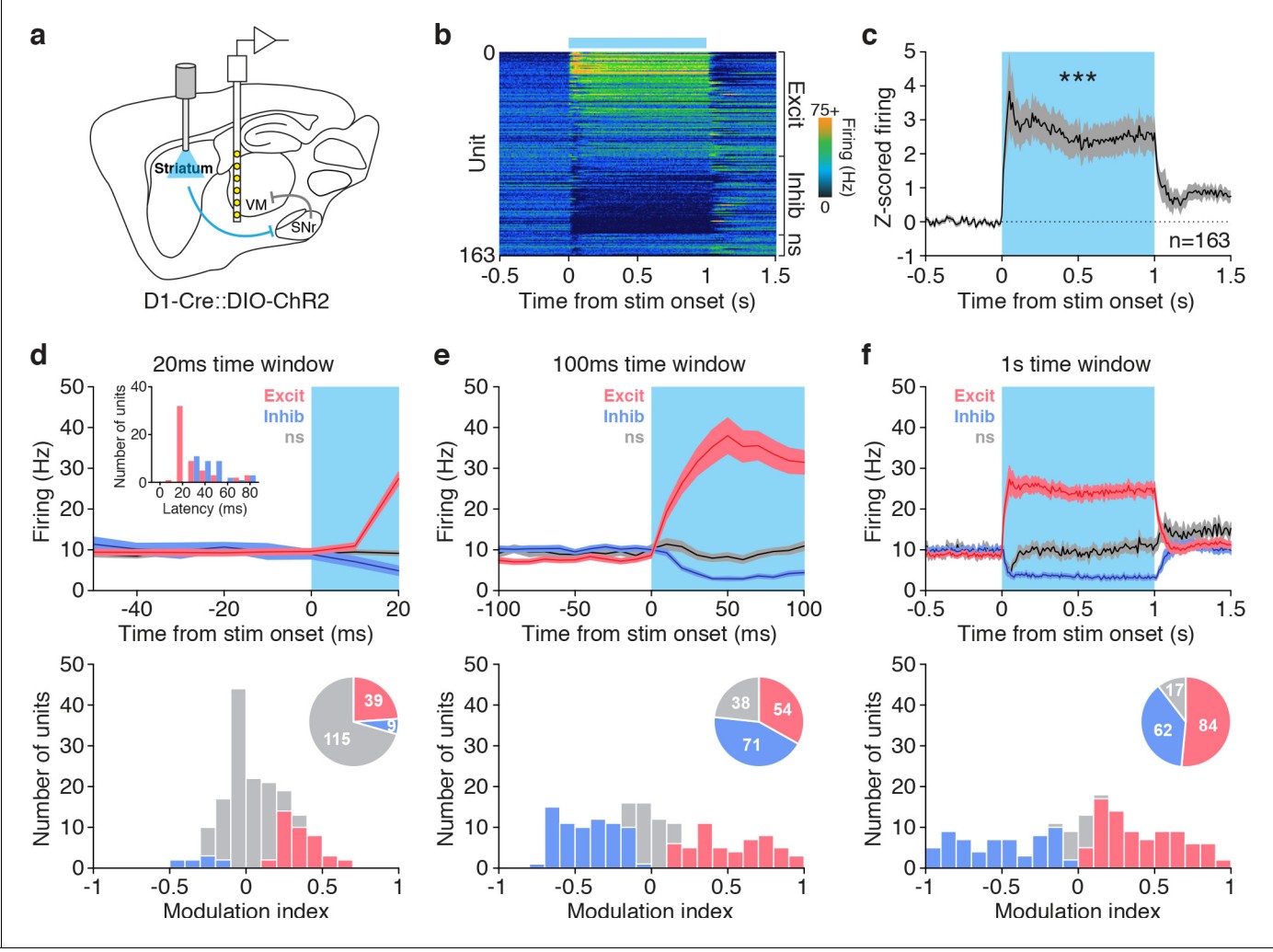

**Figure 6.** dMSN stimulation increases firing in VM thalamus. (a) Sagittal schematic depicting dMSN optogenetic stimulation with simultaneous in vivo single-unit recording from VM in awake, head-fixed mouse. (b) Peri-event time histogram of all 163 cells recorded in VM, aligned to dMSN stimulation (blue bar, 1 s duration). Units are sorted by response type (increase [excit], no significant change [ns], decrease [inhib]) and modulation index (see below). (c) Average z-scored response of all VM neurons to dMSN stimulation (blue shading, 1 s duration, n = 163, baseline vs stim, Wilcoxon signed rank test, p=0.0006). (d–f) Detailed analysis of responses for the first 20 ms (d), first 100 ms (e) or full 1 s (f) of dMSN stimulation (blue shading). Average firing rate (top) and modulation index (bottom) for excited (red), inhibited (blue) or unmodulated units (grey). Latency to significant change in firing rate is shown in d, top inset. Bottom insets, pie charts showing fraction of excited (red), inhibited (blue) and unmodulated (grey) VM neurons during dMSN stimulation for each time window. See materials and methods for modulation index calculation.

DOI: https://doi.org/10.7554/eLife.34032.021

The following source data and figure supplement are available for figure 6:

**Source data 1.** Source data for *Figure 6*.
DOI: https://doi.org/10.7554/eLife.34032.023

**Figure supplement 1.** Stimulation and recording sites for VM in vivo recordings.
DOI: https://doi.org/10.7554/eLife.34032.022

*Beutler et al., 2011*). Altogether, these results are consistent with the idea that simple forms of reinforcement may not require corticostriatal synaptic plasticity, but rather take advantage of a different mechanism.

## Multiple basal ganglia output targets for reinforcement

Several basal ganglia output targets in the brainstem have been associated with reinforcement. Recently, a series of optogenetic studies reported a role for both DRN serotonergic and non-

serotonergic populations in reinforcement. Consistent with these reports, we observed that mice self-stimulated serotonergic neurons of the DRN (*Figure 3*). We observed reinforcement in SERT-Cre mice, which failed to sustain operant behavior in a previous report (*Fonseca et al., 2015*). In ePET-Cre mice, we did not observe reinforcement, consistent with one previous report (*McDevitt et al., 2014*), but inconsistent with another recent study (*Liu et al., 2014*). These discrepancies could be explained by variable targeting of DRN subregions and stimulation parameters. Despite overall mixed observations about which exact neuronal population is sufficient for reinforcement, a common result is that DRN activation is reinforcing through direct excitation of midbrain dopamine neurons (*Liu et al., 2014*; *McDevitt et al., 2014*). In our hands, ablation of DRN SERT-positive neurons did not affect dMSN stimulation (*Figure 4*).

Similarly, the MLR receives projections from the SNr and has been associated with both locomotor control and reinforcement (*Inglis et al., 2000*; *Ryczko and Dubuc, 2013*). Both glutamatergic and cholinergic populations of the MLR have been associated with reinforcement (*Dautan et al., 2016*; *Xiao et al., 2016*; *Yoo et al., 2017*), and we have previously shown that striatal dMSNs drive locomotion by disinhibiting MLR glutamatergic neurons (*Roseberry et al., 2016*). We observed variable levels of MLR glutamatergic neuron self-stimulation at sites that reliably drove locomotion (*Figure 3*). However, dMSN-driven reinforcement did not require MLR glutamatergic neurons (*Figure 4*). Similar to DRN, MLR self-stimulation of both glutamatergic and cholinergic populations seems to rely on direct excitation of midbrain dopamine neurons (*Xiao et al., 2016*; *Yoo et al., 2017*). Altogether, these results show that dMSN self-stimulation does not require DRN serotonergic or MLR glutamatergic neurons, suggesting that these nuclei belong to separate reinforcement circuits.

## A role for motor thalamus in reinforcement

VM thalamus is traditionally classified as a motor nucleus relaying basal ganglia signals toward motor cortex (*Starr and Summerhayes, 1983*; *Klockgether et al., 1985*; *Turner and Desmurget, 2010*). VM and VA/VL nuclei of the thalamus receive basal ganglia outputs (*Carter and Fibiger, 1978*; *Di Chiara et al., 1979*) and project to motor cortical regions (*Herkenham, 1979*). However, VM differs with unique, widespread axonal projections to Layer I across many cortical areas including prefrontal and parietal cortices that are associated with more cognitive functions (*Kuramoto et al., 2015*).

Consistent with the disinhibitory architecture of the basal ganglia (*Deniau and Chevalier, 1985*; *Albin et al., 1989*), we observed an overall increase in VM activity upon dMSN stimulation (*Figure 6*). Latency to excitation was short (~10–20 ms), in line with di-synaptic disinhibition via the SNr. However, a significant fraction of cells showed a decrease in firing, highlighting a degree of heterogeneity among VM neurons. These differences in response type could be based on connectivity or biophysical properties of VM neurons. Alternatively, inhibition could arise from polysynaptic network effects like recruitment of inhibition (via the reticular thalamic nucleus) by VM excited neurons or through lateral disinhibition within the SNr among VM-projecting cells. Both scenarios would be consistent with the longer latencies to inhibition observed here. Nevertheless, our results demonstrate that dMSN stimulation significantly increases activity in the majority of VM neurons (*Figure 6*), which is critical for dMSN-driven reinforcement (see below).

Mimicking dMSN-driven VM disinhibition with optogenetic VM activation supported operant behavior, similar to dMSN self-stimulation and SNr self-inhibition (*Figure 3*), and consistent with VM electrical self-stimulation in rats (*Clavier and Gerfen, 1982*). In contrast to results obtained with silencing of brainstem SNr targets, both extended silencing of VM with muscimol infusion, or temporally-precise SNr terminal excitation time-locked to dMSN stimulation, decreased the total number of nosepokes mice performed to obtain dMSN stimulation (*Figure 5*). The latter approach argues for an acute role of VM disinhibition in reinforcement, rather than a global alteration of brain state. Therefore, it is possible that the magnitude of reinforcement (number of nosepokes) scales with that of VM disinhibition, similar to graded dMSN self-stimulation observed with increasing stimulation frequencies (*Figure 1—figure supplement 2*). Importantly, VM silencing did not affect either spontaneous locomotion in the open field or dMSN locomotor drive, consistent with a primary role of MLR glutamatergic output in locomotor control. Altogether, these results identify VM as a key recipient of basal ganglia reinforcement signals.

Silencing VM did not fully block dMSN self-stimulation. This may be due to incomplete VM silencing, or the participation of additional mechanisms. For example, mediodorsal thalamus, a basal

ganglia-recipient, non-motor nucleus, has been implicated in action-outcome learning (*Chudasama et al., 2001*; *Corbit et al., 2002*; *Corbit et al., 2003*) and working memory (*Parnaudeau et al., 2013*; *Bolkan et al., 2017*), and could support striatal reinforcement. The SNr also targets the superior colliculus, which displays reward-related activity (*Jeljeli et al., 2003*; *Doya, 2007*), and MLR cholinergic population, which has been shown to drive reinforcement (*Xiao et al., 2016*). Additionally, dMSN activation, through inhibition of SNc-projecting SNr neurons (*Mailly et al., 2001*; *Brazhnik et al., 2008*), may disinhibit SNc dopamine neurons and thus increase DA release, which is sufficient for reinforcement (*Phillips et al., 1976*; *Gratton et al., 1988*; *Ilango et al., 2014*). Interestingly, rodent VM is embedded in circuits separate from dopamine (*Groenewegen, 1988*; *Papadopoulos and Parnavelas, 1990*; *Watabe-Uchida et al., 2012*), suggesting that the basal ganglia could engage both dopamine-dependent and dopamine-independent reinforcement mechanisms. In fact, previous reports have shown that dopamine-deficient mice can still learn to locate food in a T-maze (*Robinson et al., 2005*) or develop place preference for morphine or cocaine (*Hnasko et al., 2005*; *Hnasko et al., 2007*), providing evidence that alternate, dopamine-independent reinforcement mechanisms can occur.

VM lesions have been shown to impair both the acquisition and consolidation of active avoidance (*Zis et al., 1984*), a behavior that is believed to depend on the basal ganglia (*Delacour et al., 1977*; *Hormigo et al., 2016*). In songbirds, vocal learning requires an intact basal ganglia-thalamocortical loop (*Scharff and Nottebohm, 1991*; *Fee and Goldberg, 2011*). Other lesion studies in rodents have suggested a role for VM in spatial navigation (*Jeljeli et al., 2003*) and working memory (*Bailey and Mair, 2005*), and more specifically in sustaining cortical motor preparatory activity in a working memory task (*Guo et al., 2017*). Indeed, VM/VAL activation and inhibition increases and decreases cortical activity, respectively. Thus, it appears that basal-ganglia-recipient motor thalamus may be important for engaging cortical plasticity and learning.

## Distinct circuits for locomotion and reinforcement

Gross manipulations of striatum, including cocaine administration and bulk dMSN stimulation drive both locomotion and reinforcement (*Di Chiara et al., 1979*; *Inglis et al., 2000*; *Kravitz et al., 2010*; *Kravitz et al., 2012*). However, it has not been clear whether those two distinct processes can be dissociated in downstream circuitry. Interestingly, direct stimulation of both VM thalamus and MLR glutamatergic neurons, two main targets of basal ganglia outputs, increases locomotion (*Klockgether et al., 1986*; *Roseberry et al., 2016*) and sustains operant responding (*Clavier and Gerfen, 1982*; *Yoo et al., 2017*). However, we demonstrate a dissociation of these two basal-ganglia-driven functions at the level of output targets: reinforcement relies on VM but not MLR glutamatergic neurons (*Figures 4* and *5*), whereas basal ganglia-driven locomotion depends on MLR glutamatergic neurons, but not VM (*Figure 5*, *Figure 4—figure supplement 1*). Future studies will be required to understand the exact nature of basal ganglia signals transferred to motor thalamus and how this information is integrated in thalamocortical circuits to support cognitive behavior.

# Materials and methods

## Subjects

All procedures were in accordance with protocols approved by the UCSF Institutional Animal Care and Use Committee. Mice were maintained on a 12 hr/12 hr light/dark cycle and fed ad libitum. Experiments were carried out during the light cycle. 166 wild-type ($C_{57}$BL/6) or transgenic (gene name in italic) mice were used for experiments as follows: 44 D1-Cre mice (*Drd1*, 27 females, 17 males, GENSAT #030778-UCD) were used for dMSN self-stimulation experiments. four male D1-Cre mice were used for in vivo recordings in VM. three female D1-Cre mice were used for dMSN-driven locomotion. 30 D1-Cre mice crossed to D1-tmt mice (Jackson #016204) and 9 D1-tmt mice were used for slice recordings in the striatum. 7 D1-Cre x NR1f/f mice (*Grin1*<sup>tm2Stl</sup>, six females, one male, Jackson #005246) were used for dMSN self-stimulation. 17 vGAT-Cre mice (*Slc32a1*, nine females, eight males, Jackson #016962) were used for SNr self-inhibition. 12 vGAT-Cre (six females, six males) were used for dMSN axon self-stimulation paired with SNr terminal stimulation in VM. one male vGAT-Cre mouse was used for slice recordings. 16 wild-type mice (15 females, one male) were used VM self-stimulation. 5 vGLUT2-Cre mice (*Slc17a6*, two females, three males, Jackon 016963) were

used for MLR self-stimulation. 18 Sert-Cre (*Slc6a4*, seven females, 11 males, Jackson 014554) mice were used for DRN self-stimulation and dMSN self-stimulation combined with DRN caspase lesions.

## Surgical procedures

All surgeries were carried out in aseptic conditions while mice were anaesthetized with isoflurane (5% for induction, 0.5–1.5% afterward) and then placed in a stereotactic frame (Kopf). The scalp was opened and bilateral holes were drilled in the skull. In DMS, viral constructs were injected through a 33-gauge steel cannula (Plastics1) using a syringe pump (World Precision Instruments) at a rate of 100–200 nl/minute. In the SNr and MLR, viral constructs were injected through a 33-gauge needle on a 5 µL NanoFil syringe (WPI), mounted on a microsyringe pump (UMP3; WPI) and controller (Micro4; WPI), at a rate of 50–100 nl/minute. Needles were removed 5 min after the injection finished. The following volumes were injected: 500–1000 nl (DMS), 250 nl (SNr), 100 nl (VM), 500 nl (MLR, DRN, SNc). The following coordinates were used for viral injections and optic fiber implantation (AP, lateral, DV virus, DV fiber, in mm, from bregma and brain surface): DMS +0.6, 1.5, 2.5, 2; SNr −3.5, 1.4, 4; VM −1.3, 0.8, 3.7, 3.2; MLR −0.8 (from lambda), 1.2, 3.6, 3.2; fiber for dMSN axon stimulation (cerebral peduncle) −1.7, 2.75, 4.0, with 14-degree lateral angle; DRN −4.1, 0.3 (right hemisphere only), 3.3, 2.8; SNc −3.7, 1.5, 4.2, 3.4. For SNr terminal stimulation in VM, VM coordinates were used for fiber implantation. After viral injection, 200-µm-diameter optical fibers (Thorlabs #FT200UMT) glued into 1.25 mm ferrules (Thorlabs CFLC128-10) were lowered through the same holes, unless noted otherwise. Ferrules were then secured to the skull with a layer of dental adhesive (C and B Metabond, Parkell), and covered in UV-cure dental cement (flowable composite, Henry Schein) solidified with brief (20 s) UV light exposure (dental LED light lamp, MUW), while the mouse's eyes were protected with cardboard glasses. The scalp was then sutured shut around the dental cement. Buprenorphine HCl (0.1 mg/kg, intraperitoneal injection) and Ketoprofen (5 mg/kg, subcutaneous injection) were used for postoperative analgesia. Mice were given 2–4 weeks for recovery and viral expression after surgery before behavioral training started.

## Viral constructs

For Cre-dependent neuronal excitation and inhibition, we used an adeno-associated virus serotype 5 (AAV5) carrying Channelrhodopsin-2 (hChR2(H124R)) or eArch3.0, respectively, fused to enhanced yellow fluorescent protein (eYFP) in a double-floxed inverted open reading frame (DIO) under the control of the EF1α promoter (AAV5-EF1α-DIO-hChR2(H124R)-eYFP, AAV5-EF1α-DIO-eArch3.0-eYFP and AAV5-EF1α-DIO-eYFP for controls, University of Pennsylvania Vector Core, diluted 1:3 in saline (ChR2 and eYFP)). For MLR glutamatergic neurons inhibition, we expressed halorhodopsin 3.0 (eNpHR3.0) under the control of CaMKIIα (AAV5-CaMKIIα-eNpHR3.0-eYFP, University of North Carolina Vector Core). For VM excitation, we expressed Channelrhodopsin-2 under the control of CaMKIIα (AAV5-CaMKIIα-hChR2(H124R)-eYFP, University of North Carolina Vector Core). For DRN serotonergic neuron ablation, we expressed Caspase three in a flip-excision switch (AAV5-FLEx-EF1α-taCasp3-TEVp, University of North Carolina Vector Core). DIO constructs were allowed at least 2 weeks before experiments, and CamKIIα constructs at least 4 weeks.

## Awake infusions in VM

For awake drug infusions in VM, mice were implanted with a custom stainless steel headbar for head fixation. The scalp was removed and skull scraped clean and dry using a scalpel. VM sites (AP, lateral, in mm, from bregma: −1.3, 0.8) were marked with sharpie pen on the skull and covered with a drop of silicone elastomer (Kwik-Cast; WPI). Cyanoacrylate glue (Vetbond, 3M) was lightly dabbed on the skull. For combined VM infusions and dMSN self-stimulation, holes above DMS were then drilled, the viral construct was injected and optic fibers implanted in DMS, and secured as described above. Then, the headbar was levelled flat and lowered to touch lambda, covered with dental adhesive (C and B Metabond, Parkell) and secured with UV-cure dental cement. Only a thin layer of was applied above the marked VM sites. After at least 2 weeks of recovery, mice were habituated to head fixation in a 3-cm-wide acrylic cylinder for 10 min twice a day, 3 days prior to the beginning of the experiment. On the last day of habituation, mice were anesthetized, dental cement and silicone above VM were removed, and holes were drilled on the marked VM locations. The craniotomy was then covered in silicone elastomere and mice were given 24–48 hr for recovery. For drug infusions,

mice were headfixed, levelled flat, and a glass pipette mounted on a microinjector (Nanoject) was lowered 3.7 mm from brain surface through the craniotomy to deliver 100 nl of muscimol (0.1 mM in saline, Tocris) over 2 min in VM. The pipette was removed 2 min later and the same procedure was repeated on the other hemisphere. The craniotomy was covered in silicone elastomer and mice were immediately transferred to the behavior boxes for training. After the last behavior session, mice were infused with 20 nl of DiI (ThermoFischer) on each side and then perfused.

## Behavioral experiments

### Self-Stimulation

#### Apparatus

The self-stimulation apparatus consisted of a 10 × 20 cm box made of white acrylic, equipped with two nosepokes with infrared beams (MIX-engineering) on the same wall and a houselight (MedAssociates). For optogenetic excitation, we used 473 nm wavelength lasers (Shanghai Lasers) adjusted to 1 mW at the output of the fiber optic implant for continuous exposure or 10 mW for pulsed excitation (Master-8, A.M.P.I). For optogenetic inhibition, we used continuous 532 nm wavelength adjusted to 3–10 mW at the output of the fiber optic implant. Laser-emitted light was fed in 200 μm, 0.39NA optic fiber cable (Thorlabs), split through a 1-to-2 commutator (Doric) to two optic fiber cables connected to the mouse's head for bilateral neuronal manipulation. Behavior was performed in the dark. Mouse position and nosepokes were tracked with Noldus software (Ethovision).

#### Self-stimulation

Sessions lasted 30 min. The night before the first training session, mice were food- and water-restricted. For the first training session, both nosepokes were baited with 200 nl of 10% sucrose-containing water, to ensure apparatus exploration. Mice were bilaterally connected to the laser with fiber optic cables and placed in the box. Poking in the laser-paired nosepoke (active) triggered the laser and the houselight, and poking in the other nosepoke (inactive) had no effect. The nosepoke-laser contingency was counterbalanced across mice and fixed throughout training. Mice could reinitiate a nosepoke as soon as the laser turned off, or after a similar lapse for the inactive side, by exiting and reentering the poke. After the first session, mice went back to unlimited water and food access for the rest of the training. Across all conditions, total number of nosepokes was stable from the first session onward. Mice were excluded if by day four the number of active nosepokes had declined by more than 50% or if mice performed <1 poke/min on average over four sessions. Laser activation patterns and durations were set as follows: dMSN cell bodies, 1 s continuous; dMSN axons, 1 s 40 Hz, SNr, 3 s continuous; VM, 3 s 20 Hz; MLR excitation 3 s 20 Hz; MLR inhibition, 1.1 s continuous, with dMSN stimulation starting 0.1 s later, for 1 s; SNr terminals in VM, 1 s 20 Hz, simultaneous to dMSN axon stimulation; DRN 3 s 40 Hz; SNc 1 s 40 Hz. Pulse duration, 5 ms.

Throughout the manuscript, self-stimulation data come from the first behavior session, except for *Figure 1c,d,h,i*, where data come from the first session with a 1 s laser/nosepoke contingency (second overall session).

The extinction session happened 24 hr after the 4th training session. Mice were placed in the apparatus for 15 min, during which the previously laser-paired nosepoke became inactive. Mice intended for ex vivo recordings were trained for 4 days and slices were prepared 24 hr later.

#### Locomotion

Spontaneous and dMSN-driven locomotion was measured in a 20 cm diameter circular open field arena made of transparent acrylic placed on a square of white yoga mat. Mice were habituated for 30 min to the open field 1 day prior to the experiment. After VM infusions, mice were placed in the open field for 5 min to measure baseline locomotion. For dMSN-driven locomotion, we repeated five cycles consisting of 10 s pre-stimulation, 10 s of bilateral optogenetic stimulation and 10 s post-stimulation epochs, followed by 30 s of recovery. Each mouse received both saline and muscimol infusions in VM in a random order, 24 hr apart. For dMSN stimulation paired with MLR glutamate neurons inhibition, the same protocol was applied, except that dMSN stimulation alone cycles were alternated with dMSN stimulation paired with bilateral MLR inhibition cycles, 5 cycles of each. Total distance travelled was summed for each epoch type, and averaged across mice.

## Electrophysiology in acute slices

Mice were euthanized with a lethal dose of ketamine and xylazine followed by transcardial perfusion with 8 ml of ice cold artificial corticospinal fluid (aCSF) containing (in mM): glycerol (250), KCl (2.5), $MgCl_2$ (2), CaCl2 (2), NaH2PO4 (1.2), HEPES (10), $NaHCO_3$ (21) and glucose (5). Coronal slices (250 µM) containing the DMS or the SNr were then prepared with a vibratome (Leica) in the same solution, before incubation in 33°C recording aCSF containing (in mM): NaCl (125), $NaHCO_3$ (26), $NaH_2PO_4$ (1.25), KCl (2.5), $MgCl_2$ (1), $CaCl_2$ (2), glucose (12.5), continuously bubbled with 95/5% $O_2/CO_2$. After 30 min of recovery, slices were either kept at room temperature or transferred to a recording chamber superfused with recording aCSF (2.5 ml/min) at 33°C. Whole-cell current-clamp recordings were obtained using an internal solution containing (in mM): KGluconate (130), NaCl (10), $MgCl_2$ (2), $CaCl_2$ (0.16), EGTA (0.5), HEPES (10). Voltage-clamp recordings were obtained using an internal solution containing (in mM): $CsMeSO_3$ (120), CsCl (15), NaCl (8), EGTA (0.5), HEPES (10), Mg-ATP (2), Na-GTP (0.3), TEA-Cl (10), QX-314 (5). EPSCs were evoked with a monopolar stimulation electrode (glass pipet) placed in the striatum, between the recorded cell and the cortex, and isolated in presence of picrotoxin (100 µM). TTX (1 µM) was added to isolate miniEPSCs. Cells were held at −70 mV except for NMDAR currents, which were recorded at +40 mV (measured 50 ms after stimulus onset). dMSNs were identified by tomato fluorescence. ChR2-positive neurons were identified by the presence of a photocurrent evoked by 500 ms blue light. eYFP-positive neurons were identified by eYFP fluorescence. SNr terminal excitation was achieved by flashing 470 nm filtered LED (Thorlabs LED4C driven by a Prizmatix BLCC-2) light through a 40x immersion objective (1–5 ms pulse, 0.1–1 mW/cm$^2$). Holding current was varied from −70 to +40 mV. In VM, currents evoked from SNr terminal stimulation were only observed at positive potential and were blocked by picrotoxin (100 µM). Stimulation was applied every 10 s. Holding currents were not corrected for junction potential. Recordings were obtained with 3–4 MW resistance pipettes pulled from glass capillaries (Harvard Apparatus GC150TF-10) on a puller (Zeitz). Picrotoxin (Tocris) was prepared at stock concentration in $H_2O$, then diluted in aCSF while kynurenic acid (Sigma) was directly added to aCSF for bath application. Data was acquired with custom written code in Igor software, which can be found in the source code files linked to this article.

## In vivo electrophysiology

D1-Cre mice were injected with DIO-ChR2 virus, implanted with optical fibers over DMS, and implanted with headbars as described above. Recordings were performed in head-fixed mice sitting in an acrylic tube. Mice were acclimated to head-fixation in several habituation sessions during the week before recording. On the day of recording, craniotomies (500 µm diameter) were performed over VM (0.7 mm posterior of Bregma, 0.8 lateral of midline) in the left and right hemispheres. The dura was left intact. Craniotomies were covered with a drop of surgilube followed by silicon elastomer and the animal was allowed to recover for at least 4 hr. Acute recordings were performed using a 64 channel silicon probe (H2 Cambridge Neurotech) mounted on a micromanipulator. The probe was inserted into the craniotomy and driven vertically to a depth of ~4 mm. To reduce movement, the craniotomy was covered with 2% agarose in saline and a layer of mineral oil. The probe was allowed to settle for at least 15 min before acquiring data. Continuous broadband data were acquired at 30 kHz using a SpikeGadgets acquisition system and Trodes software. Blue light was delivered to the DMS ipsilateral to the recording site using a fiber-coupled LED (M470F3, Thorlabs). 1 s of stimulation was delivered every 10 s for 100 trials. The power at the fiber tip was 1 mW. Probe shanks were coated with CM-DiI.

### Analysis

Spike sorting was performed using MountainSort software (*Chung et al., 2017*). After a manual merging step, clusters with a clear refractory period in their autocorrelogram, noise overlap metric <0.03 (*Chung et al., 2017*), and amplitude distribution without a sharp cutoff at low amplitudes were considered single units. Only units whose peak signal was recorded on an electrode site within VM were included in the analysis. We report data from 163 units recorded across 8 hemispheres of 4 mice. Peri-event time histograms (PETH) aligned to the onset of stimulation were computed for each unit's firing rate using 10 ms bins. Z-scored PETHs were computed using the average and standard deviation of the PETH in a 1000 ms baseline period preceding stimulus onset.

To determine whether a unit's firing was significantly modulated by the stimulation, a paired t-test was performed to compare the firing rate in a 20 ms, 100 ms, or 1000 ms response window following stimulus onset with the firing rate in a 1000 ms baseline window preceding stimulus onset in each trial. Units with p<0.05 were considered significantly modulated for a given response window. Significantly modulated units were defined as excited if their average firing rate during the response window was greater than the baseline and inhibited if it was less. The modulation index was calculated as:

(AvgFiringRespWindow-AvgFiringBaselineWindow) / (AvgFiringRespWindow + AvgFiringBaselineWindow)

The latency to significant modulation was defined as the second time bin in which two consecutive bins of the PETH were outside of the 97% confidence level of the poisson distribution calculated from the average firing rate 1000 ms prior to stimulus onset (*Abeles, 1982*). Latency was not computed for units lacking two consecutive bins outside of the 97% confidence level within the first 100 ms after stimulus onset.

### Histology

After recordings were completed, animals were prepared for histology as described below. To identify recording sites, 100 μm coronal sections were cut on a freezing microtome and DiI signal from the electrode was identified using fluorescence microscopy. Location of electrode sites relative to VM were determined by matching the histological images to the Paxinos mouse atlas.

## Histology

Animals were euthanized with a lethal dose of ketamine and xylazine (400 mg ketamine plus 20 mg xylazine per kilogram of body weight, i.p.) and transcardially perfused with PBS, followed by 4% paraformaldehyde (PFA). Following perfusion, brains were transferred into 4% PFA for 16–24 hr and then moved to a 30% sucrose solution in PBS for 2–3 d (all at 4 deg C). Brains were then frozen and cut into 30 μm coronal sections with a sliding microtome (Leica Microsystems, model SM2000R) equipped with a freezing stage (Physitemp). Free-floating slides were blocked for 1 hr in 10% Normal Donkey Serum (NDS) in 0.5% phosphate-buffer saline in Tween 20 (PBST) then incubated overnight in primary antibody (1:500; Aves chicken anti-YFP, #GFP-1020; Pel-Freez rabbit anti-TH, P40101; ImmunoStar goat anti-5HT, 20079), 3% NDS in 0.5% PBST. The following day, they were washed 3 times for 10 min each in 0.5% PBST and incubated for 1 hr in secondary antibody (1:1000, Jackson ImmunoResearch Donkey anti-Chicken 488, #703-545-155; Invitrogen Alexa-fluor donkey anti-rabbit 647, A-31573; Invitrogen Alexa-fluor donkey anti goat 647, A21447), 3% NDS in 0.5% PBST. Slices were then incubated for 5 min with 1:2000 DAPI. After this, slides were washed for 10 min in 0.5% PBST and 2 more 10 min periods with 1:1 PBS. Slides were then washed with 0.05% lithium carbonate and alcohol, rinsed with diH2O, mounted and coverslipped with Cytoseal 60.

Slides were scanned on a VS120 semi-automated fluorescent slide scanner (Olympus Scientific Solutions Americas Corp, USA). Some figure images were acquired using a 6D high throughput microscope (Nikon, USA). All images shown were made brighter for better print quality using Photoshop function 'Vibrance', 'Brightness' and 'Contrast' to change LUT. No detail was lost during this manipulation. Manual registration of slices was performed by scaling whole slice images in Adobe Illustrator, using the Paxinos mouse atlas (Academic Press, Orlando, FL) panels as a background reference.

## Statistics

All statistics were calculated with Prism 7 (Graphpad) or Igor Pro (WaveMetrics). Distribution normality was tested for each data set to determine whether parametric or non-parametric tests would be adequate. Based on the number of groups and independent variables, we used t-test, Mann-Whitney U-test, 1-way and 2-way ANOVAs, followed by posthoc tests correcting for multiple comparisons. Only two-tailed tests were used. Tests and p-values are mentioned in figure legends. p-value<0.05 was considered significant (p<0.05*, p<0.01**, p<0.001***). Results are reported as mean +- SEM. Error bars represent SEM. When possible, sample size was calculated based on pilot cohort (powerandsamplesize.com) with a power of 0.8 and alpha of 0.05. All behavioral experiments

were independently repeated at least twice. All data supporting the findings can be found in the source data file linked to this article.

## Acknowledgements

We thank D Schulte and B Margolin for technical assistance. We thank J Berke and members of the Kreitzer lab for useful discussions. This research was supported by the Swiss National Science Foundation (ALL) and the NIH (ACK).

## Additional information

### Funding

| Funder | Grant reference number | Author |
|---|---|---|
| Schweizerischer Nationalfonds zur Förderung der Wissenschaftlichen Forschung | | Arnaud L Lalive |
| National Institutes of Health | U01 NS094342 | Anatol C Kreitzer |
| National Institutes of Health | P01 DA010154 | Anatol C Kreitzer |
| National Institutes of Health | R01 NS064984 | Anatol C Kreitzer |

The funders had no role in study design, data collection and interpretation, or the decision to submit the work for publication.

### Author contributions

Arnaud L Lalive, Conceptualization, Resources, Formal analysis, Supervision, Funding acquisition, Validation, Investigation, Visualization, Writing—original draft, Project administration, Writing—review and editing; Anthony D Lien, Christopher H Donahue, Investigation, Methodology; Thomas K Roseberry, Conceptualization, Formal analysis, Funding acquisition, Validation, Investigation, Visualization, Writing—original draft, Project administration, Writing—review and editing; Anatol C Kreitzer, Conceptualization, Resources, Supervision, Funding acquisition, Investigation, Methodology, Writing—original draft, Writing—review and editing

### Author ORCIDs

Anatol C Kreitzer  http://orcid.org/0000-0001-7423-2398

### Ethics

Animal experimentation: This study was performed in strict accordance with the recommendations in the Guide for the Care and Use of Laboratory Animals of the National Institutes of Health. All of the animals were handled according to approved institutional animal care and use committee (IACUC) protocols (AN144957) of the University of California, San Francisco. All surgery was performed under isoflurane anesthesia, and every effort was made to minimize suffering.

### Decision letter and Author response

Decision letter https://doi.org/10.7554/eLife.34032.026
Author response https://doi.org/10.7554/eLife.34032.027

## Additional files

### Supplementary files

• Transparent reporting form
DOI: https://doi.org/10.7554/eLife.34032.024

### Data availability

All data generated or analysed during this study are included in the manuscript and supporting files.

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
