## [Decision Letter]

Thank you for submitting your article "Motor Thalamus Supports Striatum-Driven Reinforcement" for consideration by *eLife*. Your article has been reviewed by Michael Frank as the Senior Editor and Reviewing Editor, and three reviewers. The following individual involved in review of your submission has agreed to reveal his identity: D James Surmeier (Reviewer #1).

The reviewers have discussed the reviews with one another and the Reviewing Editor has drafted this decision to help you prepare a revised submission.

Summary:

This study uses optogenetic approaches to assess the role of circuitry downstream to the basal ganglia (BG) for mediating reinforcement. First, the authors demonstrate that stimulation of direct pathway medium spiny neurons (dMSN) is highly reinforcing but ostensibly not dopamine dependent. They moved on to directly stimulate BG output nucleus, substantia nigra reticulata (SNr), and find that its direct inhibition is also mildly reinforcing. SNr projection targets, including thalamus and brainstem were also directly stimulated to assess if they can support reinforcement. Similar to some previous reports, the authors find that stimulation at these sites is indeed reinforcing. Interestingly, this study also suggests that thalamic, but not brainstem, nuclei is directly involved in reinforcement produced following dMSN stimulation.

Essential revisions:

All involved found the study to be highly interesting and provocative. However, they also expressed a few major reservations that would need to be addressed.

1) There was particular concern among all three reviewers about the claims regarding dopamine independence. Their respective comments on this issue are pasted below, but the key point is that to establish this properly would require much more extensive experiments (as well as other statistics, see comments from reviewer 3) than possible for a revision at *eLife*. Nevertheless, we all agreed that the dopamine-independence is not critical for the key contribution of this study, which is mainly to show that the thalamic output is at least partially involved in dMSN-driven reinforcement. Hence, we suggest that you strongly temper the conclusions regarding DA-independence.

*Reviewer 1*:

It is not clear that the reinforcing effects of dMSN stimulation are independent of dopamine (DA). Neural networks, like neurons, are non-linear. The implicit assumption made by the authors is that partial inhibition of DA signaling should result in partial inhibition of dMSN stimulation induced behavior. I don't see any compelling reason to believe this is true. Given the novelty of the conclusions drawn by the authors, the burden of proof is on their shoulders. The author's assertion is based upon two negative observations: nosepokes were not affected by (1) cocktail of a D1R (SCH23390) and D2R (eticlopride) antagonists administered i.p. and (2) deletion of NR1 subunits from dMSNs. The D1R/D2R cocktail produced a very modest reduction in motor behavior and in self-stimulation of SNc. For this to be convincing, there must be no 'spare' receptors in the circuitry mediating the behavior. Second, not all of the effects of DA on reinforcement need to be mediated by NMDARs – DA receptors also change the intrinsic excitability of basal ganglia neurons; only long-term memory of reinforcing events is thought require long-term potentiation of glutamatergic synapses and NMDARs (behavior over days was examined (Results section) but not discussed; why?). At the very least, the authors should determine how dMSN optical stimulation interacts with forms of reinforcement that are well known to depend upon DA. For example, does nosepoking for dMSN stimulation affect nose-poking for sucrose? Previous work by Tai et al., and Yttri and Dudman suggests it should, implicating DA signaling.

DA-dependent synaptic plasticity may be necessary for long-term memories associated with goal directed behavior, but not necessarily the short-term memory examined in this study. It would be worthwhile testing animals on successive days of training to determine if there is in fact a memory created by dMSN stimulation.

*Reviewer 2:*

- The DA independence of D1MSN stimulation is corroborated by the dMSN-NR1-KO mice. Indeed, while the authors have reported this before (Kravitz 2012), and cite that study for DA antagonist injections, they do not explicitly indicate the route of infusions (i.e IP, intracranial) and whether it is in the direct vicinity of the optical fibers. I suspect this was an IP infusion as in the 2012 study, but it would be interesting to see the authors clearly, and directly demonstrate the DA independence of this effect. That is, without concentration confounds (movement related) and localization to str (systemic vs local).

- VM may instantiate some increased propensity of re-doing an action (similar to 'activation' role of DA) as part of the intrinsic reinforcing nature of ICSS. But, whether this develops across days or its similarity to learning that requires striatal LTP can only be clarified if that authors analyze and show effects of multi-day stimulations and other behavioral metrics in addition to total # of nosepokes.

*Reviewer 3:*

Statistical power is an issue throughout the paper, particularly when claiming

- Positive effects with an n of 2 (MLR. This after also throwing out over half of the data because it didn't fit), or

- Lack of effect (antagonists, receptor KO's had no effect). Eye-balling the data in Figure 2g (antagonist), I roughly get saline control (u= 325, std= 165) and antag (u=235, std = 95), with n = 7 in each group.

First, failure to detect an effect *does not* mean the two distributions are equivalent or that there is no effect. This is, sadly, a common mistake and to say so requires a different statistical test.

Second, any measure of power makes it clear that, with this small sample size and large variance, it would be highly likely to commit a type 2 error – failure to detect a difference that is there. From this estimation, the authors would need to double their sample size to get even 70% power.

In fact, although it is difficult to ascertain, it appears that be increasing power by combining this result with the data from 2A of the author's 2010 paper (albeit with a slightly different antagonist), the effect of antagonists is significant, rather than merely a strong trend across multiple papers.

This, in combination with the smallest concentration of eticlopride I've seen used (the standard for awake behaving rodents is at least 0.5 mg/kg, not 0.2) casts some doubt on this, their first major finding.

2) Also, from multiple reviewers: The electrophysiological consequences of optical stimulation of dMSNs on downstream targets are not systematically examined but simply inferred. The paper is about the circuitry mediating the effects of dMSN stimulation. Yet, outside of a few recordings from VM neurons, there is no attempt to determine the electrophysiological consequences of optogenetic stimulation on key circuit elements. This also is true in other parts of the study (e.g., SNr terminal stimulation is assumed to have a simple effect on VM neurons). More work needs to be done characterizing the thalamic response. Characterizing an area with 13 neurons, and making broad claims with 6 of those 13, is troubling. In supplemental Figure 5 there are very modest differences between excitation and inhibition of VM neurons by dMSN stimulation. In fact, excitation won by only 3 cells out of 13! Parametric Z-scores cannot be used to assay population behavior with a sample size of 13. The experiment illustrated in Figure 4F-H is elegant, but the nose-pokes evoked by terminal stimulation are twice those evoked by striatal stimulation in the previous figures. This makes it difficult to evaluate. Moreover, what is to be made of the change in slope, rather than the shift in curves seen in panel B?

Moreover, the questions the authors could answer would greatly and easily strengthen the impact of the paper – not only to give a better sense of the thalamic response (there appears to be evidence of excitatory responses early, middle and late follow stim – but it's impossible to tell with such a small sample size), but to get an idea how the thalamus responds during behavior. The one recording was done in a head-fixed mouse that passively received stimulation. Answering anything along the lines of how do the neurons respond during unstimulated nose pokes, how does the response change over a session, and how/why do the two populations of early excite and early inhib+late excite come about would be wonderful to see. I think this falls in line with the earlier comment concerning the assumption of simple linearity. That a brain region responds when another is stimulated says almost nothing about its potential to modulate learning or behavior. In my opinion, this could be done in a week with a few animals, taking the recording from troubling weakness to outstanding strength.

3) The experiments performed do not exclude a role for the MLR in mediating the effects of dMSN stimulation. None of the experiments address the potential role of cholinergic or GABAergic neurons in this region. The cholinergic neurons here have widespread projections, including to the thalamus. Moreover, it is not clear that the optogenetic inhibition of glutamatergic neurons in this region included those projecting to the SNc and basal ganglia. So, the assertion that the brainstem circuits don't play a role needs to be qualified, or else the cholinergic inputs studies.

4) In Figure 1D, dMSN stimulation routinely promotes ~400 pokes/30 minute session (ignoring a couple of outliers), and Figure 2 indicates that the direct stimulation of projection targets individually contribute ~ 100-200 pokes/ 30 minutes. Finally, disruption of Vm with muscimol or activation of GABAergic input from SNr reduce nosepoke responding to ~150 pokes/30 minutes. Altogether these results suggest a capacity of DRN and MLR for reinforcement, but the authors claim that "dMSN self stimulation does not engage DRN or MLR for reinforcement". What structure, then, is responsible for the remaining reinforcement when SNr-> VM is activated coincideRt-pcrnt with dMSN activation. In other words, VM thalamus modulates the reinforcing effects of dMSN stimulation, though inhibition only blunts, not eliminates the effect. This is potentially and due to very interesting non-VM mechanisms, and it would be a great addition for the authors to address it.

5) Situating these results in context of a previous extensive ICSS literature. The reinforcing effects of ICSS have been shown to occur throughout the brain, including cerebellum, PFC, LC, Raphe, brainstem (including putative MLR), thalamus – and under the control of either DA, Ach, 5HT, and/or noradrenaline. ICSS 'reinforced' behavior had been studied in depth in the late 60's and 70's and continues to today, with particularly excellent work done in the early 70's by Routtenberg et al. The absence of any citation of this extensive literature, save for one focusing on the BG and dopamine by Phillips, (1976), is problematic given that this paper focuses on ICSS (even though it does study a more specific component related to dMSN stimulation and downstream targets).

Given this, I would like the authors to read some of the seminal work on ICSS throughout the brain in order to better understand the context (that most positive ICSS is observed outside of the BG and is DA-independent was common knowledge in 1975). Wise, 1996 would be a wonderful start, and his list of positive ICSS locations is below. The authors re-discovery of non-DA ICSS everywhere is not exciting – it happens even in primary sensory cortex. However, their dissociation and mapping of the pathways that execute striatum's reinforcing and motor roles is certainly valuable. I would recommend a reworking of the manuscript to take this into account.

The authors' overcoming "the requirement for dopamine in reinforcement (Discussion section)" is a terrible straw man, one that the authors identify as already having been torn down in the very next paragraph. Phillips himself suggested that ICSS may be intrinsically reinforcing, rather than being a part of a brain's native RL mechanism.

From Wise, 1996 (Figure 1): I. medial forebrain bundle sites, including Ihe anterior, posterior, and lateral hypothalamus; 2. ventromedial hypothalamus; 3. substantia nigra (zona compacta) and ventral tegmental area;4. midline mesencephalon, including the regions of the dorsal and medial raphe nuclei; 5. region of locus coeruleus; 6. deep cerebellar nuclei and decussation of the brachium conjunctivum; 7. regions of the mesencephalic and motor nuclei of the trigeminal nerve; 8. nucleus of the solitary tract; 9. olfactory bulb; 10. olfactory tubercle; 11. medial frontal cortex; 12. sulcal frontal cortex; 13. anterior cingulate cortex; 14. entorhinal cortex; 15. hippocampus; 16. amygdala; 17. medial and lateral septal regions; 18. nucleus accumbens; 19. caudate nucleus: and 20. dorso-medial thalamus.

---

## [Author Response]

All involved found the study to be highly interesting and provocative. However, they also expressed a few major reservations that would need to be addressed.1) There was particular concern among all threRt-pcre reviewers about the claims regarding dopamine independence. Their respective comments on this issue are pasted below, but the key point is that to establish this properly would require much more extensive experiments (as well as other statistics, see comments from reviewer 3) than possible for a revision at eLife. Nevertheless, we all agreed that the dopamine-independence is not critical for the key contribution of this study, which is mainly to show that the thalamic output is at least partially involved in dMSN-driven reinforcement. Hence, we suggest that you strongly temper the conclusions regarding DA-independence.

We have taken the reviewers’ comments into account and decided to remove the DA antagonist experiment from the manuscript. Accordingly, we no longer draw any specific conclusions regarding the role of DA in dMSN-driven reinforcement and have rewritten sections across the whole manuscript. Instead, we use the data presented in Figure 1 to introduce the hypothesis of a downstream circuit mechanism for reinforcement.

Reviewer 1:

It is not clear that the reinforcing effects of dMSN stimulation are independent of dopamine (DA). Neural networks, like neurons, are non-linear. The implicit assumption made by the authors is that partial inhibition of DA signaling should result in partial inhibition of dMSN stimulation induced behavior. I don't see any compelling reason to believe this is true. Given the novelty of the conclusions drawn by the authors, the burden of proof is on their shoulders. The author's assertion is based upon two negative observations: nosepokes were not affected by (1) cocktail of a D1R (SCH23390) and D2R (eticlopride) antagonists administered i.p. and (2) deletion of NR1 subunits from dMSNs. The D1R/D2R cocktail produced a very modest reduction in motor behavior and in self-stimulation of SNc. For this to be convincing, there must be no 'spare' receptors in the circuitry mediating the behavior. Second, not all of the effects of DA on reinforcement need to be mediated by NMDARs – DA receptors also change the intrinsic excitability of basal ganglia neurons; only long-term memory of reinforcing events is thought require long-term potentiation of glutamatergic synapses and NMDARs (behavior over days was examined (Results section) but not discussed; why?). At the very least, the authors should determine how dMSN optical stimulation interacts with forms of reinforcement that are well known to depend upon DA. For example, does nosepoking for dMSN stimulation affect nose-poking for sucrose? Previous work by Tai et al., and Yttri and Dudman suggests it should, implicating DA signaling.

The reviewer raises a number of valid points. We agree that much more work would be required to convincingly demonstrate DA independence. We have therefore removed the DA antagonist experiments from the manuscript.

DA-dependent synaptic plasticity may be necessary for long-term memories associated with goal directed behavior, but not necessarily the short-term memory examined in this study. It would be worthwhile testing animals on successive days of training to determine if there is in fact a memory created by dMSN stimulation.

We have now added data into Figure 1 showing that our protocol does elicit a long-term (>24 hour) memory of the previously-paired nosepoke. We also added new data showing that despite this learning, no observable changes are observed in mEPSC frequency, mEPSC amplitude, paired-pulse ratio, AMPA/NMDA ratio, or intrinsic excitability in stimulated dMSNs recorded in slices after training (Figure 2) or in neighboring iMSNs (Figure 2—figure supplemRt-pcrent 1).

Reviewer 2:

- The DA independence of D1MSN stimulation is corroborated by the dMSN-NR1-KO mice. Indeed, while the authors have reported this before (Kravitz 2012), and cite that study for DA antagonist injections, they do not explicitly indicate the route of infusions (i.e IP, intracranial) and whether it is in the direct vicinity of the optical fibers. I susRt-pcrpect this was an IP infusion as in the 2012 study, but it would be interesting to see the authors clearly, and directly demonstrate the DA independence of this effect. That is, without concentration confounds (movement related) and localization to str (systemic vs local).

As described above, we have decided to remove the DA antagonist experiments from the manuscript.

- VM may instantiate some increased propensity of re-doing an action (similar to 'activation' role of DA) as part of the intrinsic reinforcing nature of ICSS. But, whether this develops across days or its similarity to learning that requires striatal LTP can only be clarified if that authors analyze and show effects of multi-day stimulations and other behavioral metrics in addition to total # of nosepokes.

We now show dMSN self-stimulation over multiple days (Figure 1—figure supplement 3). We analyzed multiple aspects of the self-stimulation (poking patterns, interpoke intervals) and found that behavior on consecutive days was similar to the first session. This suggests that behavior was learned and stabilized within a single session.

Reviewer 3:

Statistical power is an issue throughout the paper, particularly when claiming- Positive effects with an n of 2 (MLR. This after also throwing out over half of the data because it didn't fit), or- Lack of effect (antagonists, receptor KO's had no effect). Eye-balling the data in Figure 2g (antagonist), I roughly get saline control (u= 325, std= 165) and antag (u=235, std = 95), with n = 7 in each group.First, failure to detect an effect *does not* mean the two distributions are equivalent or that there is no effect. This is, sadly, a common mistake and to say so requires a different statistical test.Second, any measure of power makes it clear that, with this small sample size and large variance, it would be highly likely to commit a type 2 error – failure to detect a difference that is there. From this estimation, the authors would need to double their sample size to get even 70% power.In fact, although it is difficult to ascertain, it appears that be increasing power by combining this result with the data from 2A of the author's 2010 paper (albeit with a slightly different antagonist), the effect of antagonists is significant, rather than merely a strong trend across multiple papers.This, in combination with the smallest concentration of eticlopride I've seen used (the standard for awake behaving rodents is at least 0.5 mg/kg, not 0.2) casts some doubt on this, their first major finding.

We agree that we do not present a strong case for dopamine independence and thus we have removed these results from the manuscript and toned down these conclusions.

Regarding MLR glutamatergic neurons self-stimulation, we have removed the statistical test and explain here the classification of responders versus non-rRt-pcresponders. Our aim was to show that these neurons, which control locomotion and mediate dMSNdriven locomotion (Roseberry et al., 2016, and our manuscript, Figure 4—figure supplement 1), can also, in some instances, elicit reinforcement. Indeed, all 5 mice showed an increase in locomotion upon stimulation (data not shown). However, only 2 of them consistently self-stimulated (responsive). The other 3 were classified as non-responsive (<1 poke/minute on average over 4 sessions or >50% decrease by session 4, a selection criterion applied to all conditions across the paper. This criterion is mentioned in the ‘transparent reporting’ file, submitted along with the manuscript, and can now be found in the methods section). Given that at least some of these mice showed reinforcement seems sufficient to support further investigation of the role of MLR glutamatergic neurons in dMSN-driven reinforcement (which is the point of this Figure).

2) Also, from multiple reviewers: The electrophysiological consequences of optical stimulation of dMSNs on downstream targets are not systematically examined but simply inferred. The paper is about the circuitry mediating the effects of dMSN stimulation. Yet, outside of a few recordings from VM neurons, there is no attempt to determine the electrophysiological consequences of optogenetic stimulation on key circuit elements. This also is true in other parts of the study (e.g., SNr terminal stimulation is assumed to have a simple effect on VM neurons). More work needs to be done characterizing the thalamic response. Characterizing an area with 13 neurons, and making broad claims with 6 of those 13, is troubling. In supplemental Figure 5 there are very modest differences between excitation and inhibition of VM neurons by dMSN stimulation. In fact, excitation won by only 3 cells out of 13! ParameRt-pcrtric Z-scores cannot be used to assay population behavior with a sample size of 13. The experiment illustrated in Figure 4F-H is elegant, but the nose-pokes evoked by terminal stimulation are twice those evoked by striatal stimulation in the previous figures. This makes it difficult to evaluate. Moreover, what is to be made of the change in slope, rather than the shift in curves seen in panel B?Moreover, the questions the authors could answer would greatly and easily strengthen the impact of the paper – not only to give a better sense of the thalamic response (there appears to be evidence of excitatory responses early, middle and late follow stim – but it's impossible to tell with such a small sample size), but to get an idea how the thalamus responds during behavior. The one recording was done in a head-fixed mouse that passively received stimulation. Answering anything along the lines of how do the neurons respond during unstimulated nose pokes, how does the response change over a session, and how/why do the two populations of early excite and early inhib+late excite come about would be wonderful to see. I think this falls in line with the earlier comment concerning the assumption of simple linearity. That a brain region responds when another is stimulated says almost nothing about its potential to modulate learning or behavior. In my opinion, this could be done in a week with a few animals, taking the recording from troubling weakness to outstanding strength.

We performed additional in vivo recordings and gathered data from 163 neurons to characterize VM activity during dMSN stimulation in head-fixed, awake mice. To better mimic behavior, we recorded VM responses during a 1second stimulation window (versus only 100ms in the previous version). These results are now presented in Figure 6. After 20ms of dMSN stimulation, most responsive VM neurons were excited. After 100ms of stimulation, a significant fraction of neurons with decreased activity appeared. After 1second, a majority of neurons were excited. Overall, these results fit with our model of dMSN-driven VM disinhibition for reinforcement. Additionally, we now discuss the heterogeneous response properties of VM neurons during dMSN stimulation. We believe head-fixed recordings are not only optimal but also necessary to properly isolate VM responses during dMSN stimulation from any movement-related activity. Given the known role of VM in motor control, recording from freely-moving animals is highly likely to confound dMSN-driven activity with spontaneous movement related activity. In order to further link specific VM activity patterns with reinforcement versus movement, future studies should take advantage of head-fixed operant tasks combined with in vivo recordings. Although extremely interesting, we believe that these experiments fall beyond the scope of our manuscript.

Regarding the experiments shown in Figure 5f-H (previously Figure 4F-H):

although dMSN axonal self-stimulation was stronger than cell body self-stimulation, its combination with SNr terminal excitation in VM led to an approximate 50% decrease, which is similar to the muscimol effect on cell body self-stimulation shown in Figure 5.

Regarding Figure 5B and G, we apologize that the chosen examples did not accurately reflect the average behavior. We do observe a change in slope with both pharmacological and optogenetic silencing approaches, which fits with a decrease in poking rate / total number of nosepokes. We replaced this example with another trace which better represents the average result.

3) The experiments performed do not exclude a role for the MLR in mediating the effects of dMSN stimulation. None of the experiments address the potential role of cholinergic or GABAergic neurons in this region. The cholinergic neurons here have widespread projections, including to the thalamus. Moreover, it is not clear that the optogenetic inhibition of glutamatergic neurons in this region included those projecting to the SNc and basal ganglia. So, the assertion that the brainstem circuits don't play a role needs to be qualified, or else the cholinergic inputs studies.

We apologize for the lack of clarity. Our experiments are strictly limited to glutamatergic neurons of the MLR, which we functionally define here as sufficient for locomotion and necessary for dMSN-driven locomotion. We have updated the text to be more specific.

4) In Figure 1D, dMSN stimulation routinely promotes ~400 pokes/30 minute session (ignoring a couple of outliers), and Figure 2 indicates that the direct stimulation of projection targets individually contribute ~ 100-200 pokes/ 30 minutes. Finally, disruption of Vm with muscimol or activation of GABAergic input from SNr reduce nosepoke responding to ~150 pokes/30 minutes. Altogether these results suggest a capacity of DRN and MLR for reinforcement, but the authors claim that "dMSN self stimulation does not engage DRN or MLR for reinforcement". What structure, then, is responsible for the remaining reinforcement when SNr-> VM is activated coincident with dMSN activation. In other words, VM thalamus modulates the reinforcing effects of dMSN stimulation, though inhibition only blunts, not eliminates the effect. This is potentially and due to very interesting non-VM mechanisms, and it would be a great addition for the authors to address it.

Several possibilities could explain the partial decrease in self-stimulation during VM manipulation. First, it is possible that the small volume and concentration of muscimol that we used does not fully silence VM. Additionally, the SNr also projects to other regions than the ones examined in this study, like the thalamic mediodorsal nucleus, involved in learning and working memory (Parnaudeau et al., 2013). The SNr targets the superior colliculus, which displays reward-related activity (Ikeda and Hikosaka, 2003; Weldon et al., 2007), and MLR cholinergic population, which has been shown to drive reinforcement (Xiao et al., 2016). Lastly, as the reviewers have pointed out, our resRt-pcrRt-pcrRt-pcrults do not fully discard a role for DA in dMSN self-stimulation. Identifying the remaining component for dMSN-driven reinforcement is a highly interesting question, that we now discuss in the manuscript and would like to identify in future studies.

5) […] Given this, I would like the authors to read some of the seminal work on ICSS throughout the brain in order to better understand the context (that most positive ICSS is observed outside of the BG and is DA-independent was common knowledge in 1975). Wise, 1996 would be a wonderful start, and his list of positive ICSS locations is below. The authors re-discovery of non-DA ICSS everywhere is not exciting – it happens even in primary sensory cortex. However, their dissociation and mapping of the pathways that execute striatum's reinforcing and motor roles is certainly valuable. I would recommend a reworking of the manuscript to take this into account.The authors' overcoming "the requirement for dopamine in reinforcement (Discussion section)" is a terrible straw man, one that the authors identify as already having been torn down in the very next paragraph. Phillips himself suggested that ICSS may be intrinsically reinforcing, rather than being a part of a brain's native RL mechanism.From Wise, 1996 (Figure 1): I. medial forebrain bundle sites, including Ihe anterior, posterior, and lateral hypothalamus; 2. ventromedial hypothalamus; 3. substantia nigra (zona compacta) and ventral tegmental area;4. midline mesencephalon, including the regions of the dorsal and medial raphe nuclei; 5. region of locus coeruleus; 6. deep cerebellar nuclei and decussation of the brachium conjunctivum; 7. regions of the mesencephalic and motor nuclei of the trigeminal nerve; 8. nucleus of the solitary tract; 9. olfactory bulb; 10. olfactory tubercle; 11. medial frontal cortex; 12. sulcal frontal cortex; 13. anterior cingulate cortex; 14. entorhinal cortex; 15. hippocampus; 16. amygdala; 17. medial and lateral septal regions; 18. nucleus accumbens; 19. caudate nucleus: and 20. dorso-medial thalamus.

We agree and now discuss our data within the framework of past ICSS literature.